# Towards Scalable Unpaired Virtual Try-On via Patch-Routed Spatially-Adaptive GAN

**Zhenyu Xie**[1], **Zaiyu Huang**[1], **Fuwei Zhao**[1], **Haoye Dong**[2]
**Michael Kampffmeyer**[3], **Xiaodan Liang**[1,4]*
[1]Shenzhen Campus of Sun Yat-Sen University, [2]Huya Inc
[3]UiT The Arctic University of Norway, [4]Peng Cheng Laboratory
{xiezhy6,huangzy225,zhaofw}@mail2.sysu.edu.cn, donghaoye@huya.com
michael.c.kampffmeyer@uit.no, xdliang328@gmail.com

## Abstract

Image-based virtual try-on is one of the most promising applications of human-centric image generation due to its tremendous real-world potential. Yet, as most try-on approaches fit in-shop garments onto a target person, they require the laborious and restrictive construction of a paired training dataset, severely limiting their scalability. While a few recent works attempt to transfer garments directly from one person to another, alleviating the need to collect paired datasets, their performance is impacted by the lack of paired (supervised) information. In particular, disentangling style and spatial information of the garment becomes a challenge, which existing methods either address by requiring auxiliary data or extensive online optimization procedures, thereby still inhibiting their scalability. To achieve a *scalable* virtual try-on system that can transfer arbitrary garments between a source and a target person in an unsupervised manner, we thus propose a texture-preserving end-to-end network, the PAtch-routed SpaTially-Adaptive GAN (PASTA-GAN), that facilitates real-world unpaired virtual try-on. Specifically, to disentangle the style and spatial information of each garment, PASTA-GAN consists of an innovative patch-routed disentanglement module for successfully retaining garment texture and shape characteristics. Guided by the source person keypoints, the patch-routed disentanglement module first decouples garments into normalized patches, thus eliminating the inherent spatial information of the garment, and then reconstructs the normalized patches to the warped garment complying with the target person pose. Given the warped garment, PASTA-GAN further introduces novel spatially-adaptive residual blocks that guide the generator to synthesize more realistic garment details. Extensive comparisons with paired and unpaired approaches demonstrate the superiority of PASTA-GAN, highlighting its ability to generate high-quality try-on images when faced with a large variety of garments (e.g. vests, shirts, pants), taking a crucial step towards real-world scalable try-on.

## 1 Introduction

Image-based virtual try-on, the process of computationally transferring a garment onto a particular person in a query image, is one of the most promising applications of human-centric image generation with the potential to revolutionize shopping experiences and reduce purchase returns. However, to fully exploit its potential, scalable solutions are required that can leverage easily accessible training data, handle arbitrary garments, and provide efficient inference results. Unfortunately, to date, most existing methods [35, 38, 12, 7, 37, 9, 10, 4, 36, 39] rely on *paired* training data, i.e., a person image

---

*Xiaodan Liang is the corresponding author. Our code will be available at PASTA-GAN.

35th Conference on Neural Information Processing Systems (NeurIPS 2021).

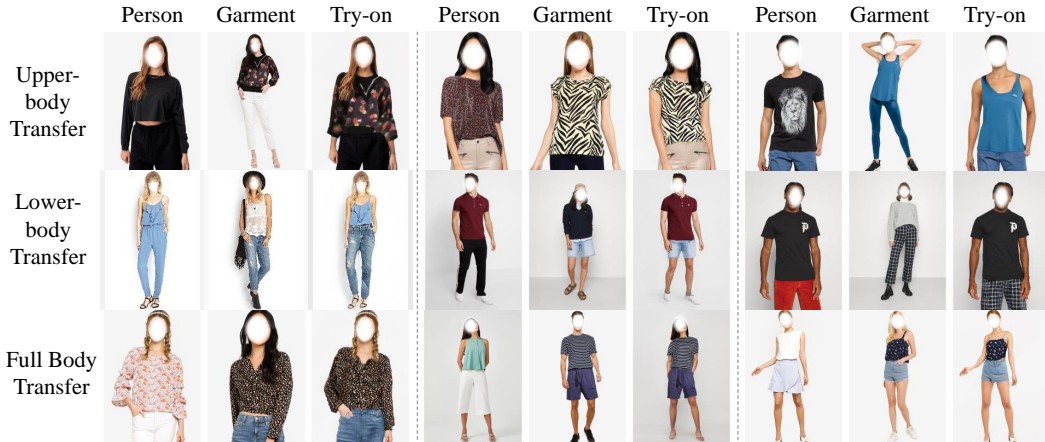

Figure 1: Example virtual try-on results from our PASTA-GAN, which is flexible for various try-on scenarios, e.g., garment transfer for the upper body, the lower body, and the full body.

and its corresponding in-shop garment, leading to laborious data-collection processes. Furthermore, these methods are unable to exchange garments directly between two person images, thus largely limiting their application scenarios and raising the need for *unpaired* solutions to ensure scalability.

While *unpaired* solutions have recently started to emerge, performing virtual try-on in an unsupervised setting is extremely challenging and tends to affect the visual quality of the try-on results. Specifically, without access to the paired data, these models are usually trained by reconstructing the same person image, which is prone to over-fitting, and thus underperform when handling garment transfer during testing. The performance discrepancy is mainly reflected in the garment synthesis results, in particular the shape and texture, which we argue is caused by the entanglement of the garment style and spatial representations in the synthesis network during the reconstruction process.

While this is not a problem for the traditional paired try-on approaches [35, 12, 37, 10], which avoid this problem and preserve the garment characteristics by utilizing a supervised warping network to obtain the warped garment in target shape, this is not possible in the unpaired setting due to the lack of warped ground truth. The few works that do attempt to achieve unpaired virtual try-on, therefore, choose to circumvent this problem by either relying on person images in various poses for feature disentanglement [23, 33, 32, 31, 1, 5], which again leads to a laborious data-collection process, or require extensive online optimization procedures [25, 17] to obtain fine-grain details of the original garments, harming the inference efficiency. However, none of the existing unpaired try-on methods consider the problem of coupled style and spatial garment information directly, which is crucial to obtain accurate garment transfer results in the unpaired and unsupervised virtual try-on scenario.

In this paper, to tackle the essential challenges mentioned above, we propose a novel PAtch-routed SpaTially-Adaptive GAN, named PASTA-GAN, a scalable solution to the unpaired try-on task. Our PASTA-GAN can precisely synthesize garment shape and style (see Fig. 1) by introducing a patch-routed disentanglement module that decouples the garment style and spatial features, as well as a novel spatially-adaptive residual module to mitigate the problem of feature misalignment.

The innovation of our PASTA-GAN includes three aspects: First, by separating the garments into normalized patches with the inherent spatial information largely reduced, the patch-routed disentanglement module encourages the style encoder to learn spatial-agnostic garment features. These features enable the synthesis network to generate images with accurate garment style regardless of varying spatial garment information. Second, given the target human pose, the normalized patches can be easily reconstructed to the warped garment complying with the target shape, without requiring a warping network or a 3D human model. Finally, the spatially-adaptive residual module extracts the warped garment feature and adaptively inpaints the region that is misaligned with the target garment shape. Thereafter, the inpainted warped garment features are embedded into the intermediate layer of the synthesis network, guiding the network to generate try-on results with realistic garment texture.

We collect a scalable UnPaired virtual Try-on (UPT) dataset and conduct extensive experiments on the UPT dataset and two existing try-on benchmark datasets (i.e., the DeepFashion [21] and the MPV [6] datasets). Experiment results demonstrate that our unsupervised PASTA-GAN outperforms both the previous unpaired and paired try-on approaches.

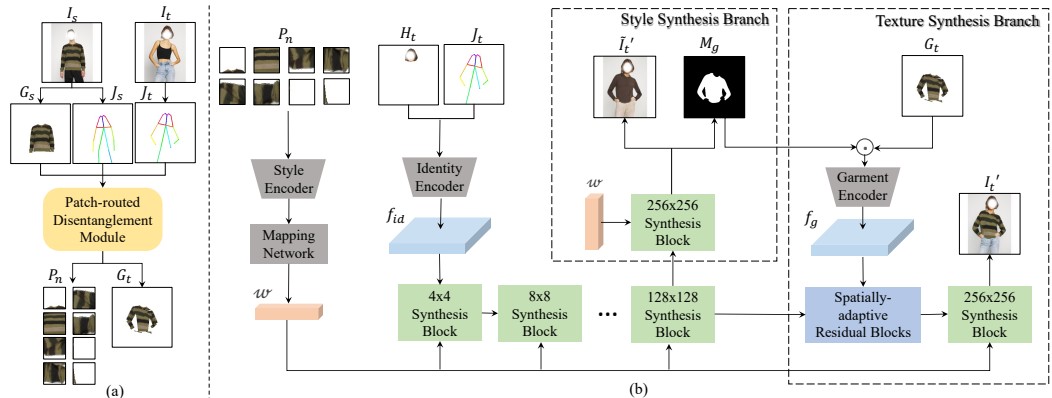

Figure 2: Overview of the inference process. (a) Given the source and target images of person $(I_s, I_t)$, we can extract the source garment $G_s$, the source pose $J_s$, and the target pose $J_t$. The three are then sent to the patch-routed disentanglement module to yield the normalized garment patches $P_n$ and the warped garment $G_t$. (b) The modified conditional StyleGAN2 first collaboratively exploits the disentangled style code $w$, projected from $P_n$, and the person identity feature $f_{id}$, encoded from target head and pose $(H_t, J_t)$, to synthesize the coarse try-on result $\tilde{I}_t'$ in the style synthesis branch along with the target garment mask $M_g$. It then leverages the warped garment feature $f_g$ in the texture synthesis branch to generate the final try-on result $I_t'$.

## 2   Related Work

**Paired Virtual Try-on.** Paired try-on methods [13, 35, 38, 12, 24, 37, 9, 10, 36] aim to transfer an in-shop garment onto a reference person. Among them, VITON [13] for the first time integrates a U-Net [29] based generation network with a TPS [2] based deformation approach to synthesize the try-on result. CP-VTON [35] improves this paradigm by replacing the time-consuming warping module with a trainable geometric matching module. VTNFP [38] adopts human parsing to guide the generation of various body parts, while [24, 37, 39] introduce a smooth constraint for the warping module to alleviate the excessive distortion in TPS warping. Besides the TPS-based warping strategy, [12, 36, 10] turn to the flow-based warping scheme which models the per-pixel deformation. Recently, VITON-HD [4] focuses on high-resolution virtual try-on and proposes an ALIAS normalization mechanism to resolve the garment misalignment. PF-AFN [10] improves the learning process by employing knowledge distillation, achieving state-of-the-art results. However, all of these methods require paired training data and are incapable of exchanging garments between two person images.

**Unpaired Virtual Try-on.** Different from the above methods, some recent works [23, 33, 32, 31, 25, 17] eliminate the need for in-shop garment images and directly transfer garments between two person images. Among them, [23, 33, 32, 31, 1, 5] leverage pose transfer as the pretext task to learn disentangled pose and appearance features for human synthesis, but require images of the same person with different poses.[2] In contrast, [25, 17] are more flexible and can be directly trained with unpaired person images. However, OVITON [25] requires online appearance optimization for each garment region during testing to maintain texture detail of the original garment. VOGUE [17] needs to separately optimize the latent codes for each person image and the interpolate coefficient for the final try-on result during testing. Therefore, existing unpaired methods require either cumbersome data collecting or extensive online optimization, extremely harming their scalability in real scenarios.

## 3   PASTA-GAN

Given a source image $I_s$ of a person wearing a garment $G_s$, and a target person image $I_t$, the unpaired virtual try-on task aims to synthesize the try-on result $I_t'$ retaining the identity of $I_t$ but wearing the source garment $G_s$. To achieve this, our PASTA-GAN first utilizes the patch-routed disentanglement module (Sec. 3.1) to transform the garment $G_s$ into normalized patches $P_n$ that are mostly agnostic to the spatial features of the garment, and further deforms $P_n$ to obtain the

---

[2]As the concurrent work StylePoseGAN [31] is the most related pose transfer-based approach, we provide a more elaborate discussion of the inherent differences in the supplementary.

warped garment $G_t$ complying with the target person pose. Then, an attribute-decoupled conditional StyleGAN2 (Sec. 3.2) is designed to synthesize try-on results in a coarse-to-fine manner, where we introduce novel spatially-adaptive residual blocks (Sec. 3.3) to inject the warped garment features into the generator network for more realistic texture synthesis. The loss functions and training details will be described in Sec. 3.4. Fig. 2 illustrates the overview of the inference process for PASTA-GAN.

### 3.1 Patch-routed Disentanglement Module

Since the paired data for supervised training is unavailable for the unpaired virtual try-on task, the synthesis network has to be trained in an unsupervised manner via image reconstruction, and thus takes a person image as input and separately extracts the feature of the intact garment and the feature of the person representation to reconstruct the original person image. While such a training strategy retains the intact garment information, which is helpful for the garment reconstruction, the features of the intact garment entangle the garment style with the spatial information in the original image. This is detrimental to the garment transfer during testing. Note that the garment style here refers to the garment color and categories, i.e., long sleeve, short sleeve, etc., while the garment spatial information implies the location, the orientation, and the relative size of the garment patch in the person image, in which the first two parts are influenced by the human pose while the third part is determined by the relative camera distance to the person.

To address this issue, we explicitly divide the garment into normalized patches to remove the inherent spatial information of the garment. Taking the sleeve patch as an example, by using division and normalization, various sleeve regions from different person images can be deformed to normalized patches with the same orientation and scale. Without the guidance of the spatial information, the network is forced to learn the garment style feature to reconstruct the garment in the synthesis image.

Fig. 3 illustrates the process of obtaining normalized garment patches, which includes two main steps: (1) pose-guided garment segmentation, and (2) perspective transformation-based patch normalization. Specifically, in the first step, the source garment $G_s$ and human pose (joints) $J_s$ are firstly obtained by applying [11] and [3] to the source person $I_s$, respectively. Given the body joints, we can segment the source garment into several patches $P_s$, which can

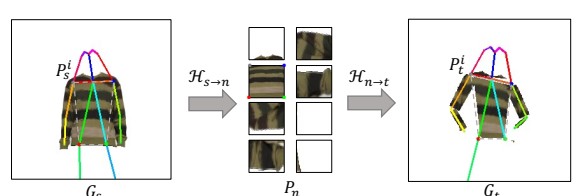

Figure 3: The process of the patch-routed deformation. Please zoom in for more details.

be quadrilaterals with arbitrary shapes (e.g., rectangle, square, trapezoid, etc.), and will later be normalized. Taking the torso region as an example, with the coordinates of the left/right shoulder joints and the left/right hips joints in $P_s^i$, a quadrilateral crop (of which the four corner points are visualized in color in $P_s^i$ of Fig. 3) covering the torso region of $G_s$ can be easily performed to produce an unnormalized garment patch. Note that we define eight patches for upper-body garments, i.e., the patches around the left/right upper/bottom arm, the patches around the left/right hips, a patch around the torso, and a patch around the neck. In the second step, all patches are normalized to remove their spatial information by perspective transformations. For this, we first define the same amount of template patches $P_n$ with fixed $64 \times 64$ resolution as transformation targets for all unnormalized source patches, and then compute a homography matrix $\mathcal{H}_{s \to n}^i \in \mathbb{R}^{3 \times 3}$ [40] for each pair of $P_s^i$ and $P_n^i$, based on the four corresponding corner points of the two patches. Concretely, $\mathcal{H}_{s \to n}^i$ serves as a perspective transformation to relate the pixel coordinates in the two patches, formulated as:

$$\begin{bmatrix} x_n^i \\ y_n^i \\ 1 \end{bmatrix} = \mathcal{H}_{s \to n}^i \begin{bmatrix} x_s^i \\ y_s^i \\ 1 \end{bmatrix} = \begin{bmatrix} h_{11}^i & h_{12}^i & h_{13}^i \\ h_{21}^i & h_{22}^i & h_{23}^i \\ h_{31}^i & h_{32}^i & h_{33}^i \end{bmatrix} \begin{bmatrix} x_s^i \\ y_s^i \\ 1 \end{bmatrix} \tag{1}$$

where $(x_n^i, y_n^i)$ and $(x_s^i, y_s^i)$ are the pixel coordinates in the normalized template patch and the unnormalized source patch, respectively. To compute the homography matrix $\mathcal{H}_{s \to n}^i$, we directly leverage the OpenCV API, which takes as inputs the corner points of the two patches and is implemented by using least-squares optimization and the Levenberg-Marquardt method [8]. After obtaining $\mathcal{H}_{s \to n}^i$, we can transform the source patch $P_s^i$ to the normalized patch $P_n^i$ according to Eq. 1.

Moreover, the normalized patches $P_n$ can further be transformed to target garment patches $P_t$ by utilizing the target pose $J_t$, which can be obtained from the target person $I_t$ via [3]. The mechanism

of that backward transformation is equivalent to the forward one in Eq. 1, i.e., computing the homography matrix $\mathcal{H}^i_{n \to t}$ based on the four point pairs extracted from the normalized patch $P^i_n$ and the target pose $J_t$. The recovered target patches $P_t$ can then be stitched to form the warped garment $G_t$ that will be sent to the texture synthesis branch in Fig. 2 to generate more realistic garment transfer results. We can also regard $\mathcal{H}_{s \to t} = \mathcal{H}_{n \to t} \cdot \mathcal{H}_{s \to n}$ as the combined deformation matrix that warps the source garment to the target person pose, bridged by an intermediate normalized patch representation that is helpful for disentangling garment styles and spatial features.

## 3.2 Attribute-decoupled Conditional StyleGAN2

Motivated by the impressive performance of StyleGAN2 [15] in the field of image synthesis, our PASTA-GAN inherits the main architecture of StyleGAN2 and modifies it to the conditional version (see Fig. 2). In the synthesis network, the normalized patches $P_n$ are projected to the style code $w$ through a style encoder followed by a mapping network, which is spatial-agnostic benefiting from the disentanglement module. In parallel, the conditional information including the target head $H_t$ and pose $J_t$ is transferred into a feature map $f_{id}$, encoding the identity of the target person by the identity encoder. Thereafter, the synthesis network starts from the identity feature map and leverages the style code as the injected vector for each synthesis block to generate the try-on result $\tilde{I}_t^{'}$.

However, the standalone conditional StyleGAN2 is insufficient to generate compelling garment details especially in the presence of complex textures or logos. For example, although the illustrated $\tilde{I}_t^{'}$ in Fig. 2 can recover accurate garment style (color and shape) given the disentangled style code $w$, it lacks the complete texture pattern. The reasons for this are twofold: First, the style encoder projects the normalized patches into a one-dimensional vector, resulting in loss of high frequency information. Second, due to the large variety of garment texture, learning the local distribution of the particular garment details is highly challenging for the basic synthesis network.

To generate more accurate garment details, instead of only having a one-way synthesis network, we intentionally split PASTA-GAN into two branches after the $128 \times 128$ synthesis block, namely the Style Synthesis Branch (SSB) and the Texture Synthesis Branch (TSB). The SSB with normal StyleGAN2 synthesis blocks aims to generate intermediate try-on results $\tilde{I}_t^{'}$ with accurate garment style and predict a precise garment mask $M_g$ that will be used by TSB. The purpose of TSB is to exploit the warped garment $G_t$, which has rich texture information to guide the synthesis path, and generate high-quality try-on results. We introduce a novel spatially-adaptive residual module specifically before the final synthesis block of the TSB, to embed the warped garment feature $f_g$ (obtained by passing $M_g$ and $G_t$ through the garment encoder) into the intermediate features and then send them to the newly designed spatialy-apaptive residual blocks, which are beneficial for successfully synthesizing texture of the final try-on result $I_t'$. The detail of this module will be described in the following section.

## 3.3 Spatially-adaptive Residual Module

Given the style code that factors out the spatial information and only keeps the style information of the garment, the style synthesis branch in Fig. 2 can accurately predict the mean color and the shape mask of the target garment. However, its inability to model the complex texture raises the need to exploit the warped garment $G_t$ to provide features that encode high-frequency texture patterns, which is in fact the motivation of the target garment reconstruction in Fig. 3.

However, as the coarse warped garment $G_t$ is directly obtained by stitching the target patches together, its shape is inaccurate and usually misaligns with the predicted mask $M_g$ (see Fig.4). Such shape misalignment in $G_t$ will consequently reduce the quality of the extracted warped garment feature $f_g$.

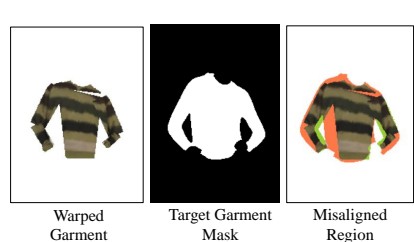

| Warped Garment | Target Garment Mask | Misaligned Region |

Figure 4: Illustration of misalignment between the warped garment and target garment shape. The orange and green region represent the region to be inpainted and to be removed, respectively.

To address this issue, we introduce the spatially-adaptive residual module between the last two synthesis blocks in the texture synthesis branch as shown in Fig. 2. This module is comprised of a garment encoder and three spatially-adaptive residual blocks with feature inpainting mechanism to modulate intermediate features by leveraging the inpainted warped garment feature.

To be specific on the feature inpainting process, we first remove the part of $G_t$ that falls outside of $M_g$ (green region in Fig.4), and explicitly inpaint the misaligned regions of the feature map within $M_g$ with average feature values (orange region in Fig. 4). The inpainted feature map can then help the final synthesis block infer reasonable texture in the inside misaligned parts.

Therefore given the predicted garment mask $M_g$, the coarse warped garment $G_t$ and its mask $M_t$, the process of feature inpainting can be formulated as:

$$M_{align} = M_g \cap M_t, \tag{2}$$

$$M_{misalign} = M_g - M_{align}, \tag{3}$$

$$f'_g = \mathcal{E}_g(G_t \odot M_g), \tag{4}$$

$$f_g = f'_g \odot (1 - M_{misalign}) + \mathcal{A}(f'_g \odot M_{align}) \odot M_{misalign}, \tag{5}$$

where $\mathcal{E}_g(\cdot)$ represents the garment encoder and $f'_g$ denotes the raw feature map of $G_t$ masked by $M_g$. $\mathcal{A}(\cdot)$ calculates the average garment features and $f_g$ is the final inpainted feature map.

Subsequently, inspired by the SPADE ResBlk from SPADE [26], the inpainted garment features are used to calculate a set of affine transformation parameters that efficiently modulate the normalized feature map within each spatially-adaptive residual block. The normalization and modulation process for a particular sample $h_{z,y,x}$ at location ($z \in C, y \in H, x \in W$) in a feature map can then be formulated as:

$$\gamma_{z,y,x}(f_g)\frac{h_{z,y,x} - \mu_z}{\sigma_z} + \beta_{z,y,x}(f_g), \tag{6}$$

where $\mu_z = \frac{1}{HW}\sum_{y,x} h_{z,y,x}$ and $\sigma_z = \sqrt{\frac{1}{HW}\sum_{y,x}(h_{z,y,x} - \mu_z)^2}$ are the mean and standard deviation of the feature map along channel $C$. $\gamma_{z,y,x}(\cdot)$ and $\beta_{z,y,x}(\cdot)$ are the convolution operations that convert the inpainted feature to affine parameters.

Eq. 6 serves as a learnable normalization layer for the spatially-adaptive residual block to better capture the statistical information of the garment feature map, thus helping the synthesis network to generate more realistic garment texture.

With the modulated intermediate feature maps produced by the spatially-adaptive residual module, the texture synthesis branch can effectively utilize the reconstructed warped garment and generate the final compelling try-on result with high-frequency texture patterns.

### 3.4 Loss Functions and Training Details

As paired training data is unavailable, our PASTA-GAN is trained unsupervised via image reconstruction. During training, we utilize the reconstruction loss $\mathcal{L}_{rec}$ and the perceptual loss [14] $\mathcal{L}_{perc}$ for both the coarse try-on result $\widetilde{I}'$ and the final try-on result $I'$:

$$\mathcal{L}_{rec} = \sum_{I \in \{\widetilde{I}', I'\}} \|I - I_s\|_1 \quad and \quad \mathcal{L}_{perc} = \sum_{I \in \{\widetilde{I}', I'\}} \sum_{k=1}^{5} \lambda_k \|\phi_k(I) - \phi_k(I_s)\|_1, \tag{7}$$

where $\phi_k(I)$ denotes the $k$-th feature map in a VGG-19 network [34] pre-trained on the ImageNet [30] dataset. We also use the $L_1$ loss between the predicted garment mask $M_g$ and the real mask $M_{gt}$ which is obtained via human parsing [11]:

$$\mathcal{L}_{mask} = \|M_g - M_{gt}\|_1. \tag{8}$$

Besides, for both $\widetilde{I}'$ and $I'$, we calculate the adversarial loss $\mathcal{L}_{GAN}$ which is the same as in Style-GAN2 [15]. The total loss can be formulated as

$$\mathcal{L} = \mathcal{L}_{GAN} + \lambda_{rec}\mathcal{L}_{rec} + \lambda_{perc}\mathcal{L}_{perc} + \lambda_{mask}\mathcal{L}_{mask}, \tag{9}$$

where $\lambda_{rec}$, $\lambda_{perc}$, and $\lambda_{mask}$ are the trade-off hyper-parameters.

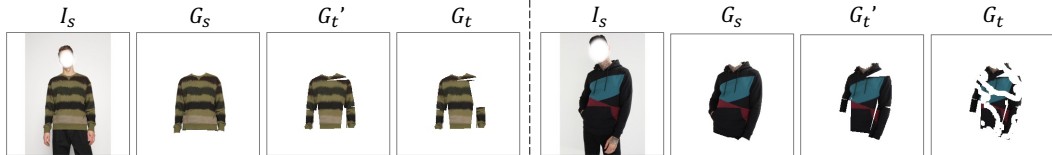

| $I_s$ | $G_s$ | $G_t{}'$ | $G_t$ | | $I_s$ | $G_s$ | $G_t{}'$ | $G_t$ |

Figure 5: Comparison among the source garment and different warped garments.

During training, although the source and target pose are the same, the coarse warped garment $G_t$ is not identical to the intact source garment $G_s$, due to the crop mechanism in the patch-routed disentanglement module. More specifically, the quadrilateral crop for $G_s$ is by design not seamless/perfect and there will accordingly often exist some small seams between adjacent patches in $G_t$ as well as incompleteness along the boundary of the torso region. To further reduce the training-test gap of the warped garment, we introduce two random erasing operations during the training phase. First, we randomly remove one of the four arm patches in the warped garment with a probability of $\alpha_1$. Second, we use the random mask from [19] to additionally erase parts of the warped garment with a probability of $\alpha_2$. Both of the erasing operations can imitate self-occlusion in the source person image. Fig. 5 illustrates the process by displaying the source garment $G_s$, the warped garment $G_t'$ that is obtained by directly stitching the warped patches together, and the warped garment $G_t$ that is sent to the network. We can observe a considerable difference between $G_t$ and $G_s$. An ablation experiment to validate the necessity of the randomly erasing operation for the unsupervised training is included in the supplementary material.

## 4 Experiments

**Datasets.** We conduct experiments on two existing virtual try-on benchmark datasets (MPV [6] dataset and DeepFashion [21] dataset) and our newly collected large-scale benchmark dataset for unpaired try-on, named UPT. UPT contains 33,254 half- and full-body front-view images of persons wearing a large variety of garments, e.g., long/short sleeve, vest, sling, pants, etc. UPT is further split into a training set of 27,139 images and a testing set of 6,115 images. In addition, we also pick out the front view images from MPV [6] and DeepFashion [21] to expand the size of our training and testing set to 54,714 and 10,493, respectively. Personally identifiable information (i.e. face information) has been masked out.

**Metrics.** We apply the Fréchet Inception Distance (FID) [27] to measure the similarity between real and synthesized images, and perform human evaluation to quantitatively evaluate the synthesis quality of different methods. For the human evaluation, we design three questionnaires corresponding to the three used datasets. In each questionnaire, we randomly select 40 try-on results generated by our PASTA-GAN and the other compared methods. Then, we invite 30 volunteers to complete the 40 tasks by choosing the most realistic try-on results. Finally, the human evaluation score is calculated as the chosen percentage for a particular method.

**Implementation Details.** Our PASTA-GAN is implemented using PyTorch [28] and is trained on 8 Tesla V100 GPUs. During training, the batch size is set to 96 and the model is trained for 4 million iterations with a learning rate of 0.002 using the Adam optimizer [16] with $\beta_1 = 0$ and $\beta_2 = 0.99$. The loss hyper-parameters $\lambda_{rec}$, $\lambda_{perc}$, and $\lambda_{mask}$ are set to 40, 40, and 100, respectively. The hhyper-parameters for the random erasing probability $\alpha_1$ and $\alpha_2$ are set to 0.2 and 0.9, respectively. [3]

**Baselines.** To validate the effectiveness of our PASTA-GAN, we compare it with the state-of-the-art methods, including three paired virtual try-on methods, CP-VTON [35], ACGPN [37], PFAFN [10], and two unpaired methods Liquid Warping GAN [20] and ADGAN [23], which have released the official code and pre-trained weights.[4] We directly use the pre-trained model of these methods as their training procedure depends on the paired data of garment-person or person-person image pairs, which are unavailable in our dataset. When testing paired methods under the unpaired try-on setting, we extract the desired garment from the person image and regard it as the in-shop garment to meet the

---

[3]Additional details for the UPT dataset (e.g., data distribution, data pre-processing), the human evaluation, training details, and the inference time analysis, etc. are provided in the supplementary material.

[4]For all these prior approaches, research use is permitted according to the respective licenses. Note, we are unable to compare with [31], [17] and [25] as they have not released their code or pre-trained model.

Table 1: The FID score [27] and human evaluation score among different methods under the unpaired setting on the DeepFashion dataset [21] and our UPT dataset.

| Method | DeepFashion | | UPT | |
|---|---|---|---|---|
| | FID ↓ | Human Evaluation ↑ | FID ↓ | Human Evaluation ↑ |
| CP-VTON [35] | 69.46 | 2.177% | 70.76 | 1.551% |
| ACGPN [37] | 44.41 | 4.597% | 37.99 | 3.448% |
| PFAFN [10] | 46.19 | 4.677% | 36.69 | 4.224% |
| ADGAN [23] | 37.36 | 21.29% | 39.60 | 7.241% |
| Liquid Warping GAN [20] | 42.18 | 12.98% | 33.18 | 9.310% |
| PASTA-GAN (Ours) | **21.58** | **54.27%** | **7.852** | **74.22%** |

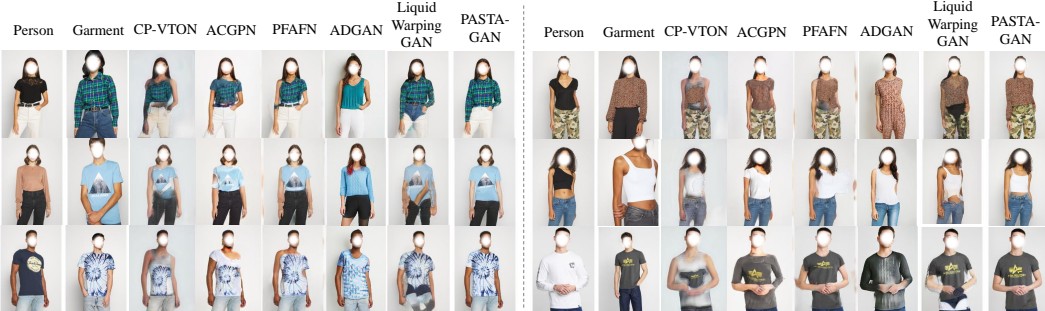

Figure 6: Visual comparison among PASTA-GAN and the baseline methods under the unpaired setting on the UPT dataset. Please zoom in for more details.

need of paired approaches. To fairly compare with the paired methods, we further conduct another experiment on the paired MPV dataset [6], in which the paired methods take an in-shop garment and a person image as inputs, while our PASTA-GAN still directly receives two person images. See the following two subsections for detailed comparisons on both paired and unpaired settings.

### 4.1 Comparison with the state-of-the-art methods on unpaired benchmark

**Quantitative:** As reported in Table 1, when testing on the DeepFashion [21] and the UPT dataset under the unpaired setting, our PASTA-GAN outperforms both the paired methods [35, 37, 10] and the unpaired methods [23, 20] by a large margin, obtaining the lowest FID score and the highest human evaluation score, demonstrating that PASTA-GAN can generate more photo-realistic images. Note that, although ADGAN [23] is trained on the DeepFashion dataset, our PASTA-GAN still surpasses it. Since the data in the DeepFashion dataset is more complicated than the data in UPT, the FID scores for the DeepFashion dataset are generally higher than the FID scores for the UPT dataset.

**Qualitative:** As shown in Fig. 6, under the unpaired setting, PASTA-GAN is capable of generating more realistic and accurate try-on results. On the one hand, paired methods [35, 37, 10] tend to fail in deforming the cropped garment to the target shape, resulting in the distorted warped garment that is largely misaligned with the target body part. On the other hand, unpaired method ADGAN [23] cannot preserve the garment texture and the person identity well due to its severe overfitting on the DeepFashion dataset. Liquid Warping GAN [20], another publicly available appearance transfer model, heavily relies on the 3D body model named SMPL [22] to obtain the appearance transfer flow. It is sensitive to the prediction accuracy of SMPL parameters, and thus prone to incorrectly transfer the appearance from other body parts (e.g., hand, lower body) into the garment region in case of inaccurate SMPL predictions. In comparison, benefited by the patch-routed mechanism, PASTA-GAN can learn appropriate garment features and predict precise garment shape. Further, the spatially-adaptive residual module can leverage the warped garment feature to guide the network to synthesize try-on results with realistic garment textures. Note that, in the top-left example of Fig. 6, our PASTA-GAN seems to smooth out the belt region. The reason for this is a parsing error. Specifically, the human parsing model [18] that was used does not designate a label for the belt, and the parsing estimator [11] will therefore assign a label for the belt region (i.e. pants, upper clothes, background, etc). For this particular example, the parsing label for the belt region is assigned the background label. This means that the pants obtained according to the predicted human parsing will

Table 2: The FID score [27] and human evaluation score among different methods under their corresponding test setting on the MPV dataset [6].

| Method | CP-VTON [35] | ACGPN [37] | PFAFN [10] | PASTA-GAN(Ours) |
|---|---|---|---|---|
| FID ↓ | 37.72 | 23.20 | 17.40 | **16.48** |
| Human Evaluation ↑ | 8.071% | 12.64% | 28.71% | **50.57%** |

Figure 7: Visual comparison among PASTA-GAN and the paired baseline methods under their corresponding test setting on the MPV dataset [6]. Please zoom in for more details.

not contain the belt, which will therefore not be contained in the normalized patches and the warped pants. The style synthesis branch then predicts the precise mask for the pants (including the belt region) and the texture synthesis branch inpaints the belt region with the white color according to the features of the pants.

## 4.2 Comparison with the state-of-the-art methods on paired benchmark

**Quantitative:** Tab. 2 illustrates the quantitative comparison on the MPV dataset [6], in which the paired methods are tested under the classical paired setting, i.e., transferring an in-shop garment onto a reference person. Our unpaired PASTA-GAN, nevertheless, can surpass the paired methods especially the state-of-the-art PFAFN [10] in both FID and human evaluation score, further evidencing the superiority of our PASTA-GAN.

**Qualitative:** Under the paired setting, the visual quality of the paired methods improves considerably, as shown in Fig. 7. The paired methods depend on TPS-based or flow-based warping architectures to deform the whole garment, which may lead to the distortion of texture and shape since the global interpolation or pixel-level correspondence is error-prone in case of large pose variation. Our PASTA-GAN, instead, warps semantic garment patches separately to alleviate the distortion and preserve the original garment texture to a larger extent. Besides, the paired methods are unable to handle garments like sling that are rarely presented in the dataset, and perform poorly on full-body images. Our PASTA-GAN instead generates compelling results even in these challenging scenarios.

## 4.3 Ablation Studies

**Patch-routed Disentanglement Module:** To validate its effectiveness, we train two PASTA-GANs without texture synthesis branch, denoted as PASTA-GAN⋆ and PASTA-GAN∗, which take the intact garment and the garment patches as input of the style encoder, respectively. As shown in Fig. 8, PASTA-GAN⋆ fails to generate accurate garment shape. In contrast, the PASTA-GAN∗ which factors out spatial information of the garment, can focus more on the garment style information, leading to the accurate synthesis of the garment shape. However, without the texture synthesis branch, both of them are unable to synthesize the detailed garment texture. The models with the texture synthesis branch can preserve the garment texture well as illustrated in Fig 8.

**Spatially-adaptive Residual Module** To validate the effectiveness of this module, we further train two PASTA-GANs with texture synthesis branch, denoted as PASTA-GAN† and PASTA-GAN‡, which excludes the style synthesis branch and replaces the spatially-adaptive residual blocks with normal residual blocks, respectively. Without the support of the corresponding components, both

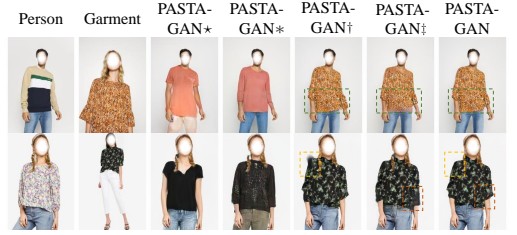

| Method | SSB | TSB | GP | NRB | SRB | FID ↓ |
|---|---|---|---|---|---|---|
| PASTA-GAN⋆ | ✓ | ✗ | ✗ | ✗ | ✗ | 21.99 |
| PASTA-GAN∗ | ✓ | ✗ | ✓ | ✗ | ✗ | 17.69 |
| PASTA-GAN† | ✗ | ✓ | ✓ | ✗ | ✓ | 11.27 |
| PASTA-GAN‡ | ✓ | ✓ | ✓ | ✓ | ✗ | 10.08 |
| PASTA-GAN | ✓ | ✓ | ✓ | ✗ | ✓ | **7.851** |

Figure 8: Qualitative results and quantitative results of the ablation study with different configurations, in which SSB, TSB, GP, NRB, SRB refer to style synthesis branch, texture synthesis branch, garment patches, normal residual blocks, and spatially-adaptive residual blocks, respectively.

PASTA-GAN† and PASTA-GAN‡ fail to fix the garment misalignment problem, leading to artifacts outside the target shape and blurred texture synthesis results. The full PASTA-GAN instead can generate try-on results with precise garment shape and texture details. The quantitative comparison results in Fig. 8 further validate the effectiveness of our designed modules.

## 5    Conclusion

We propose the PAtch-routed SpaTially-Adaptive GAN (PASTA-GAN) towards facilitating scalable unpaired virtual try-on. By utilizing the novel patch-routed disentanglement module and the spatially-adaptive residual module, PASTA-GAN effectively disentangles garment style and spatial information and generates realistic and accurate virtual-try on results without requiring auxiliary data or extensive online optimization procedures. Experiments highlight PASTA-GAN's ability to handle a large variety of garments, outperforming previous methods both in the paired and the unpaired setting.

We believe that this work will inspire new scalable approaches, facilitating the use of the large amount of available unlabeled data. However, as with most generative applications, misuse of these techniques is possible in the form of image forgeries, i.e. warping of unwanted garments with malicious intent.

## Acknowledgments and Disclosure of Funding

We would like to thank all the reviewers for their constructive comments. Our work was supported in part by National Key R&D Program of China under Grant No. 2018AAA0100300, National Natural Science Foundation of China (NSFC) under Grant No.U19A2073 and No.61976233, Guangdong Province Basic and Applied Basic Research (Regional Joint Fund-Key) Grant No.2019B1515120039, Guangdong Outstanding Youth Fund (Grant No. 2021B1515020061), Shenzhen Fundamental Research Program (Project No. RCYX20200714114642083, No. JCYJ20190807154211365), CSIG Youth Fund.

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
