# Supplementary Material for PASTA-GAN

**Zhenyu Xie**[1], **Zaiyu Huang**[1], **Fuwei Zhao**[1], **Haoye Dong**[1]
**Michael Kampffmeyer**[2], **Xiaodan Liang**[1,3*]

[1]Shenzhen Campus of Sun Yat-Sen University
[2]UiT The Arctic University of Norway, [3]Peng Cheng Laboratory
{xiezhy6,huangzy225,zhaofw,donghy7}@mail2.sysu.edu.cn
michael.c.kampffmeyer@uit.no, xdliang328@gmail.com

## 1  Architecture Details of PASTA-GAN

**Style Encoder and Mapping Network.** The Style Encoder $\mathcal{E}_S$ and mapping network $\mathcal{M}$ are used to project the garment patches into spatial-agnostic style code $w$. $\mathcal{E}_S$ is composed of one convolution layer, three encoder layers with downsampling, three encoder layers without downsampling, one average pooling layer, and one fully-connected layer. Each encoder layer consists of one dense layer and one convolution layer. $\mathcal{M}$ consists of one embedding layer and one fully-connected layer. The full architecture details can be found in Tab. 1.

**Identity Encoder.** The Identity Encoder $\mathcal{E}_I$ is used to encode the pose heat map and person head into the identity feature map $f_{id}$. It consists of eight convolution layers. More details about the architecture can be found in Tab. 2.

**Garment Encoder.** The Garment Encoder $\mathcal{E}_G$ is used to extract the warped garment feature map $f_g$, which will be used in the Spatially-adaptive Residual Blocks. It consists of a convolution layers and two Residual Blocks from [6]. More details can be found in Tab. 3.

**Generator.** The Generator $\mathcal{G}$ collaboratively exploits the garment style code $w$, identity feature map $f_{id}$, and warped garment feature map $f_g$ to generate the try-on results. It consists of six synthesis blocks (Syn. B), style synthesis branch (SSB), and texture synthesis branch (TSB). The synthesis blocks and the style synthesis branch are composed of synthesis layer (Syn. L) and RGB layer (RGB L), which are inherited from StyleGAN2 [7]. Except for the synthesis layer and RGB layer, the texture synthesis branch contains three SPADE Residual Blocks from [11]. More details about the architecture can be found in Tab. 4.

## 2  Experiments Details

### 2.1  Dataset Detail

The newly collected UPT dataset consists of front-view person images and its diversity can be summarized according to the following three aspects: First, UPT covers most of the regular garment categories, including the sling, vest, t-shirt, long sleeve shirt, coat, pants, shorts, skirts, and dress. Second, compared with the widely used virtual try-on datasets [2] and [5], UPT contains not only female images but also male images. Third, UPT contains half-body and full-body person images, with which the trained model can accomplish lower-body garment transfer and full-body garment transfer.

After collecting the raw person images, data pre-processing is required to exclude the invalid data. Specifically, we run the pose estimator [1] for each image and use the keypoint information to remove the invalid images, i.e., images without a person, images with more than one person, person images in back view, and person images that do not cover the upper body. We also apply the same filter mechanism on MPV [2] and DeepFashion [9] to obtain the front view images from these two dataset.

35th Conference on Neural Information Processing Systems (NeurIPS 2021).

Fig. 1 displays some examples of the newly collected UPT dataset and the distribution of half-body images and full-body images in the different datasets used in our experiments. Fig. 2 displays the garment category distribution in different datasets.

The images in UPT are crawled from the E-commerce website Zalando [1] and Zalora [2]. The copyright of the images belongs to these websites. We, therefore, only release the image links.

## 2.2 Training Detail

Most of the training details (e.g., data split, evaluation metrics, setting of hardware and software, setting of the optimizer, training iteration, etc.) have been described in the paper. As for the hyper-parameters of the loss functions, we first randomly choose 5% of the training data to form the validation set. Then we use the remaining training data to train the model with different hyper-parameter settings and test it on the validation set. Finally, we choose the particular hyper-parameters setting with which the trained model obtained the lowest FID score [12], and use the full training set to train our model.

## 2.3 Inference Time

We try to fairly compare the inference time among our PASTA-GAN and the other baseline methods, and find that our PASTA-GAN is superior to most of the baseline methods in terms of the inference time. More specifically, the inference time for one try-on process for CP-VTON [14], ACGPN [15], PFAFN [3], ADGAN [10], Liquid Warping GAN [8], and our proposed PASTA-GAN are 0.021s, 0.104s, 0.067s, 0.164s, 84.441s, and 0.018s, respectively. For a fair comparison, all the methods except PFAFN are tested on the same machine using one NVIDIA GeForce RTX 3090 Graphics Card. For PFAFN, we test it on another machine with one GeForce GTX 1660 Ti Graphics Card due to compatibility issues. Note that all of CP-VTON, ACGPN, ADGAN, and our PASTA-GAN rely on the 2D pose and human parsing, which requires an additional 0.190s per image (0.005s for 2D pose estimation and 0.185s for human parsing estimation). Liquid Warping GAN does not take the 2D pose and human parsing as inputs, and instead relies on the SMPL which requires much more time for the prediction. Since the official code for Liquid Warping GAN deeply entangles the data pre-processing and model inference, the inference time for Liquid Warping GAN mentioned above already includes the time for both processes. Since PFAFN leverages knowledge distillation to train a parser-free student model, there is no extra cost for data pre-processing. Finally, for one try-on process, the total inference time for CP-VTON, ACGPN, PFAFN, ADGAN, Liquid Warping GAN, and PASTA-GAN are 0.211s, 0.294s, 0.067s, 84.441s, 0.208s, respectively. We can observe that the main time cost of our PASTA-GAN is from the data pre-processing and that it has a competitive inference time compared to most of the existing virtual try-on methods.

## 2.4 Human Evaluation Details

For human evaluation, we separately design three questionnaires for the UPT dataset, the DeepFashion dataset [9], and the MPV dataset [2]. Each questionnaire is composed of 40 tasks where the volunteers need to pick out the most photo-realistic try-on results from the given options. Specifically, for each task, a person image and a garment image are provided in the question, while the virtual try-on results generated by our PASTA-GAN and the results of the other baseline methods are provided in the options in random order. The volunteers are asked to choose the synthesis result that looks most realistic and is capable of preserving the garment information, i.e., garment style and garment texture, from the garment image as much as possible. Fig. 3 shows the interface of the questionnaire for the UPT dataset. The interfaces for the other two datasets are identical.

Before the start of the human evaluation, we first invite five volunteers to accomplish the questionnaire in a serious manner to test the time required to finish an intact questionnaire. During the evaluation, for a particular questionnaire, we randomly invite 30 volunteers, who are asked to spend at least 7 seconds accomplishing each task in the questionnaire.

---

[1]https://www.zalando.co.uk/
[2]https://www.zalora.com.my/

# 3 Analysis of Limitation and Sensitivity

## 3.1 Failure Case and Limitation

In PASTA-GAN, the warped garment is derived from the garment patches of the source garment. If the garment patch for a particular body part contains some appearance from the other body parts, the warped results may be imprecise. As shown in Fig. 4(a), in the left example, the garment patch for the right arm contains the appearance of the torso, while in the right example, the garment patch for the torso contains appearance of the right arm, both of which lead to inaccurate warped results and further influence the synthesis quality of PASTA-GAN. Besides, as shown in Fig. 4(b), our PASTA-GAN fails to generate realistic try-on results when the pose of the person image is complicated and scarce in the dataset.

## 3.2 Sensitivity Analysis

To analyse the sensitivity of PASTA-GAN, we train three PASTA-GANs with different random seeds. As shown in Tab. 5, the discrepancy of the FID score [12] among the different PASTA-GANs is small, illustrating that the proposed PASTA-GAN is insensitive to the random seed.

# 4 Comparison with StylePoseGAN

There exist certain similarities in some aspects between our PASTA-GAN and the concurrent work StylePoseGAN [13], which is designed for controllable human manipulation. Specifically, both methods use a canonical garment representation and set garment feature as the input of the modulation layer in the pose-conditioned StyleGAN2 [7].

Despite this, we still have many differences inherently, summarized as follows. (1) StylePoseGAN utilizes the paired images (i.e., the same person with different poses) as training data and turns to the supervised paradigm, while our PASTA-GAN is designed for tackling the challenging unpaired try-on task where ground truth data is unavailable; (2) instead of directly using StyleGAN2 generator as a whole, we decompose the generator into a style synthesis branch and a texture synthesis branch, which separately serves to predict the precise garment mask and synthesize realistic try-on results with detailed texture; (3) our PASTA-GAN obtains the normalized garment patches by utilizing the 2D pose, while StylePoseGAN converts the garment to canonical space with the DensePose [4] UV map. Unlike the size-adjustable patch design in our PASTA-GAN, the patches obtained from DensePose can not preserve intactness of the source garment, since they only contain the texture inside the clothing-free DensePose model and thus fail in retaining the texture area outside of it. This can raise problems when dealing with loose, long sleeve garments. (4) our PASTA-GAN further transforms the normalized patches to the target shape and obtains the warped garment, which is essential for synthesizing the texture-preserved try-on results.

# 5 Additional Results

**Virtual try-on results on UPT dataset.** Fig. 5 displays more virtual try-on results generated by PASTA-GAN on the UPT dataset. The synthesis results include the upper-body transfer, lower-body transfer, and full-body transfer. Furthermore, in Fig. 6, we also display additional high resolution $(512 \times 320)$ virtual results generated by PASTA-GAN on the UPT dataset.

**Visual Comparison with the state-of-the-art methods on the UPT Dataset.** Fig. 7 displays additional visual comparisons among PASTA-GAN and the baseline methods under the unpaired setting on the UPT dataset.

**Visual comparison with the state-of-the-art methods on the DeepFashion dataset [9].** Fig. 8 displays additional visual comparisons among PASTA-GAN and the baseline methods under the unpaired setting on the DeepFashion dataset.

**Visual comparison with the state-of-the-art methods on the MPV dataset [2].** Fig. 9 displays additional visual comparisons among PASTA-GAN and the baseline methods under their corresponding setting on the MPV dataset.

**Ablation Study for Randomly Erasing Operation.** We conduct an additional ablation experiment to validate the necessity of the randomly erasing operation for the unsupervised training. More specifically, we train another PASTA-GAN (denoted as PASTA-GAN#) without conducting the randomly erasing operations on the warped garment. Thus, the warped garment that is obtained by stitching the warped patches together is directly sent to the texture synthesis branch. Then we compared PASTA-GAN# with the full PASTA-GAN both quantitatively and qualitatively. For the quantitative result, the FID score of PASTA-GAN# increases from 7.851 to 12.531 (lower is better) compared to the full PASTA-GAN model. For the qualitative results, as shown in Fig. 10, PASTA-GAN# tends to fail at synthesizing precise texture in regions which are occluded by a body part in the source person image(e.g., hair, arms, etc.). The full PASTA-GAN instead can generate realistic texture in such occluded regions.

Table 1: The architecture details of the Style Encoder $\mathcal{E}_S$ and Mapping Network $\mathcal{M}$.

| Layer | Type | Output Size |
|---|---|---|
| | $\mathcal{E}_S$ | |
| **Layer** | **Type** | **Output Size** |
| Input | Input | (64,64,30) |
| Conv | Conv2dLayer 1×1, linear | (64,64,64) |
| Enc1 | Dense Layer | (64,64,64) |
| | Conv2dLayer 3×3, down=2, linear | (32,32,128) |
| Enc2 | Dense Layer | (32,32,128) |
| | Conv2dLayer 3×3, down=2, linear | (16,16,256) |
| Enc3 | Dense Layer | (16,16,256) |
| | Conv2dLayer 3×3, down=2, linear | (8,8,512) |
| Enc4 | Dense Layer | (8,8,512) |
| | Conv2dLayer 3×3, linear | (8,8,512) |
| Enc5 | Dense Layer | (8,8,512) |
| | Conv2dLayer 3×3, linear | (8,8,512) |
| Enc6 | Dense Layer | (8,8,512) |
| | Conv2dLayer 3×3, linear | (8,8,512) |
| AVG | GlobalAveragePooling | (1,512) |
| FC1 | FullyConnectedLayer | (1,512) |
| | $\mathcal{M}$ | |
| **Layer** | **Type** | **Output Size** |
| Embed | FullyConnectedLayer | (1,512) |
| FC2 | FullyConnectedLayer | (1,512) |

Table 2: The architecture details of the Identity Encoder $\mathcal{E}_I$.

| Layer | Type | Output Size |
|---|---|---|
| | $\mathcal{E}_I$ | |
| **Layer** | **Type** | **Output Size** |
| Input | Input | (256,256,6) |
| Conv1 | Conv2dLayer 3×3, down=2, linear | (256,256,64) |
| Conv2 | Conv2dLayer 3×3, down=2, linear | (128,128,128) |
| Conv3 | Conv2dLayer 3×3, down=2, linear | (64,64,256) |
| Conv4 | Conv2dLayer 3×3, down=2, linear | (32,32,256) |
| Conv5 | Conv2dLayer 3×3, down=2, linear | (16,16,256) |
| Conv6 | Conv2dLayer 3×3, down=2, linear | (8,8,512) |

Table 3: The architecture details of the Garment Encoder $\mathcal{E}_G$.

| Layer | Type | Output Size |
|---|---|---|
| | $\mathcal{E}_G$ | |
| **Layer** | **Type** | **Output Size** |
| Input | Input | (256,256,3) |
| Conv | Conv2dLayer 7×7, relu | (256,256,64) |
| Res1 | Residual Block with conv 4×4, relu | (256,256,64) |
| Res2 | Residual Block with conv 4×4, down=2, relu | (128,128,128) |

Table 4: The architecture details of the Generator $\mathcal{G}$.

| Layer | | Type | Output Size |
|---|---|---|---|
| $\mathcal{G}$ | | | |
| Input | | Input | (4,4,512) |
| Syn. B1 | Syn. L1 | Modulated Conv 3×3, up=2, LeakyReLU | (4,4,512) |
| | RGB L1 | Modulated Conv 1×1, LeakyReLU | (4,4,3) |
| Syn. B2 | Syn. L2-1 | Modulated Conv 3×3, up=2, LeakyReLU, input from Syn. L1 | (8,8,512) |
| | Syn. L2-2 | Modulated Conv 3×3, LeakyReLU | (8,8,512) |
| | RGB L2 | Modulated Conv 1×1, LeakyReLU, skip connection from RGB L1 | (8,8,3) |
| Syn. B3 | Syn. L3-1 | Modulated Conv 3×3, up=2, LeakyReLU, input from Syn. L2-2 | (16,16,512) |
| | Syn. L3-2 | Modulated Conv 3×3, LeakyReLU | (16,16,512) |
| | RGB L3 | Modulated Conv 1×1, LeakyReLU, skip connection from RGB L2 | (16,16,3) |
| Syn. B4 | Syn. L4-1 | Modulated Conv 3×3, up=2, LeakyReLU, input from Syn. L3-2 | (32,32,512) |
| | Syn. L4-2 | Modulated Conv 3×3, LeakyReLU | (32,32,512) |
| | RGB L4 | Modulated Conv 1×1, LeakyReLU, skip connection from RGB L3 | (32,32,3) |
| Syn. B5 | Syn. L5-1 | Modulated Conv 3×3, up=2, LeakyReLU, input from Syn. L4-2 | (64,64,256) |
| | Syn. L5-2 | Modulated Conv 3×3, LeakyReLU | (64,64,256) |
| | RGB L5 | Modulated Conv 1×1, LeakyReLU, skip connection from RGB L4 | (64,64,3) |
| Syn. B6 | Syn. L6-1 | Modulated Conv 3×3, up=2, LeakyReLU, input from Syn. L5-2 | (128,128,128) |
| | Syn. L6-2 | Modulated Conv 3×3, LeakyReLU | (128,128,128) |
| | RGB L6 | Modulated Conv 1×1, LeakyReLU, skip connection from RGB L5 | (128,128,3) |
| SSB | Syn. L7-1 | Modulated Conv 3×3, up=2, LeakyReLU, input from Syn. L6-2 | (256,256,64) |
| | Syn. L7-2 | Modulated Conv 3×3, LeakyReLU | (256,256,64) |
| | RGB L7 | Modulated Conv 1×1, LeakyReLU, skip connection from RGB L6 | (256,256,4) |
| TSB | SPA. B1 | SPADE Residual Block, input from Syn. L6-2 | (128,128,128) |
| | SPA. B2 | SPADE Residual Block | (128,128,128) |
| | SPA. B3 | SPADE Residual Block | (128,128,128) |
| | Syn. L8-1 | Modulated Conv 3×3, up=2, LeakyReLU, input from SPA. B3 | (256,256,64) |
| | Syn. L8-2 | Modulated Conv 3×3, LeakyReLU | (256,256,64) |
| | RGB L8 | Modulated Conv 1×1, LeakyReLU, skip connection from RGB L6 | (256,256,3) |

Table 5: The FID score [12] of different PASTA-GAN on the testing set of UPT dataset. PASTA-GAN refers to the model used in main paper, while the other three model are the newly trained PASTA-GAN using various random seed.

| Method | PASTA-GAN | PASTA-GAN1 | PASTA-GAN2 | PASTA-GAN3 |
|---|---|---|---|---|
| FID↓ | 7.852 | 8.821 | 9.207 | 8.438 |

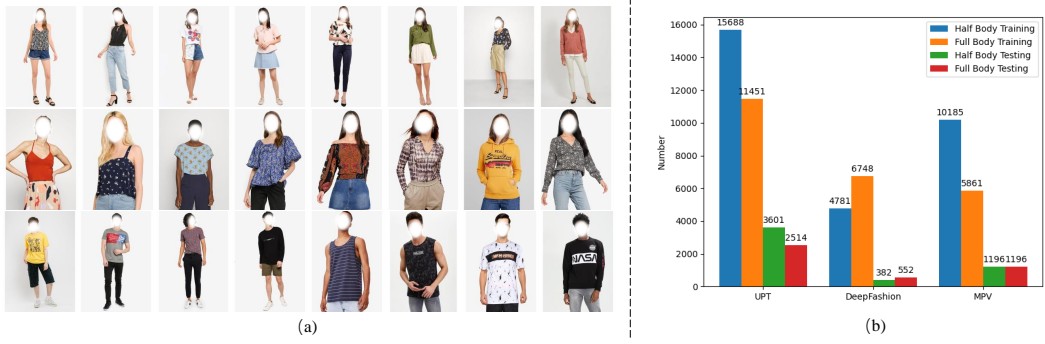

(a)            (b)

Figure 1: (a) Examples of the collected UPT dataset. (b)The distribution of the half-body images and full-body images in different datasets used in our experiments.

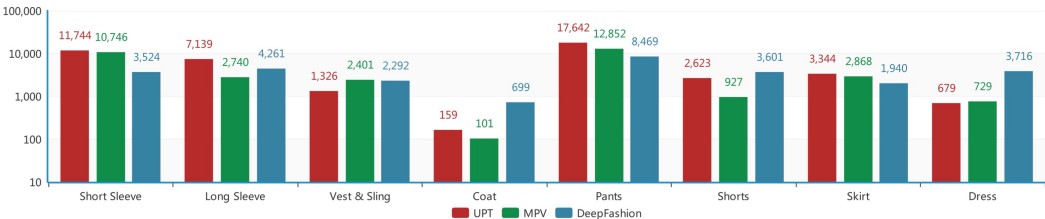

Figure 2: The distribution of the garment category in different datasets used in our experiments.

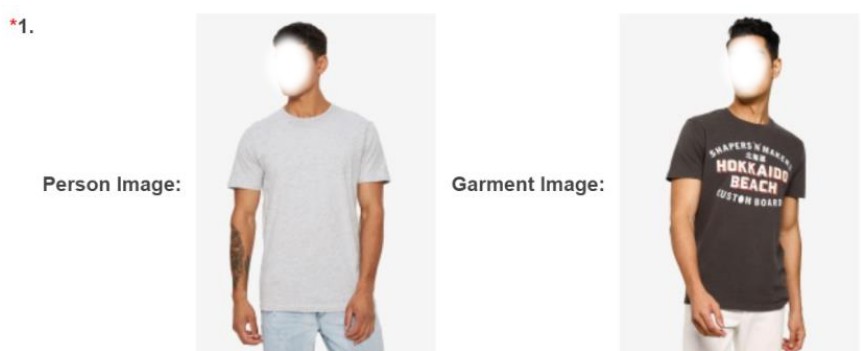

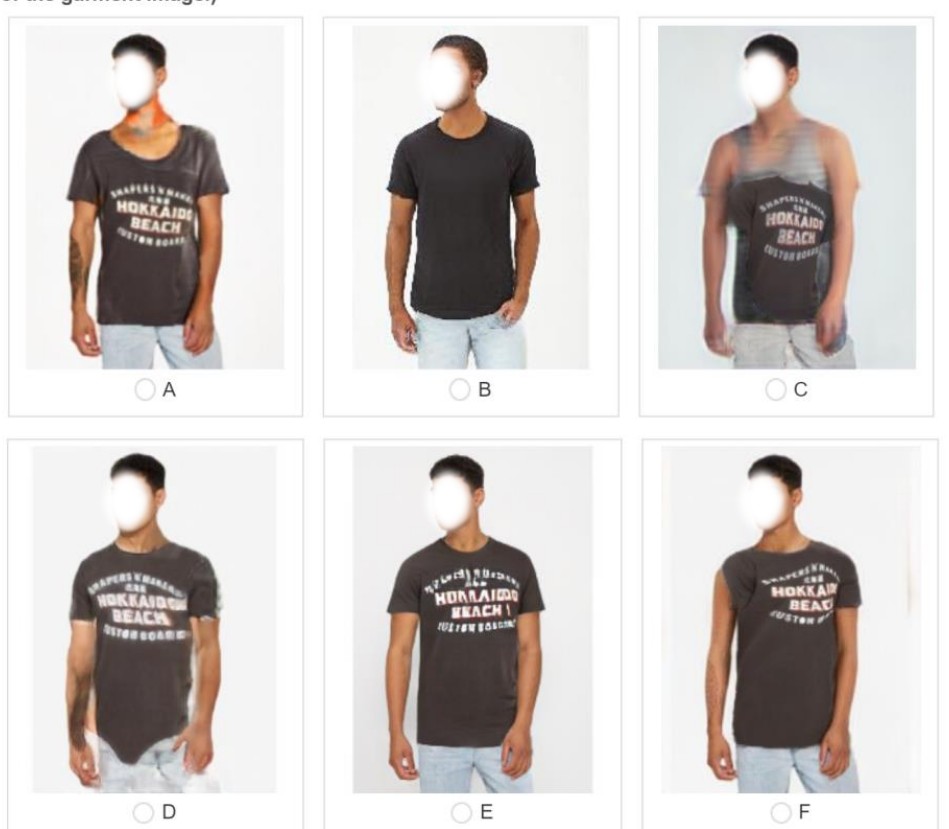

Figure 3: Interface of the task in the questionnaire for the UPT dataset.

| Person | Garment | Try-on | Person | Garment | Try-on |
|---|---|---|---|---|---|

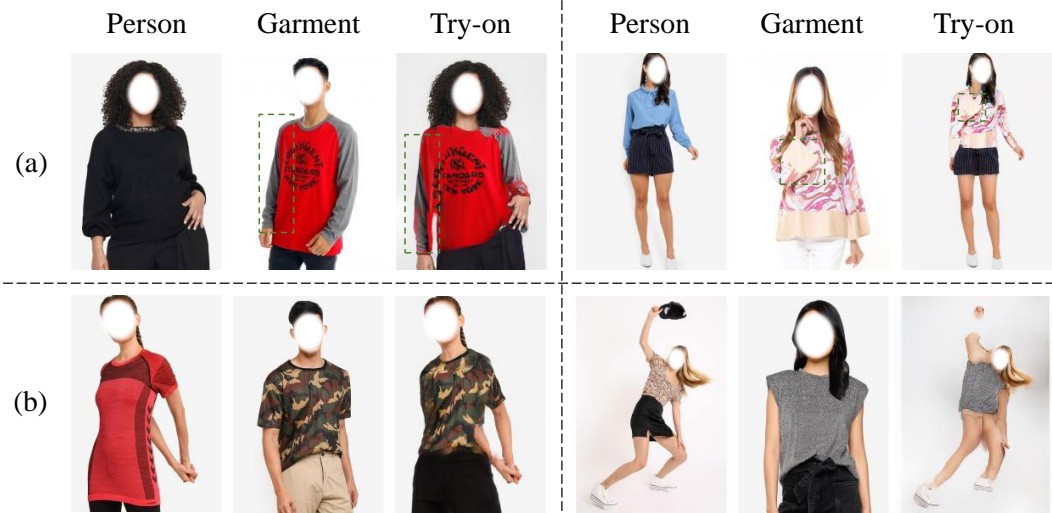

Figure 4: Failure cases of our PASTA-GAN.

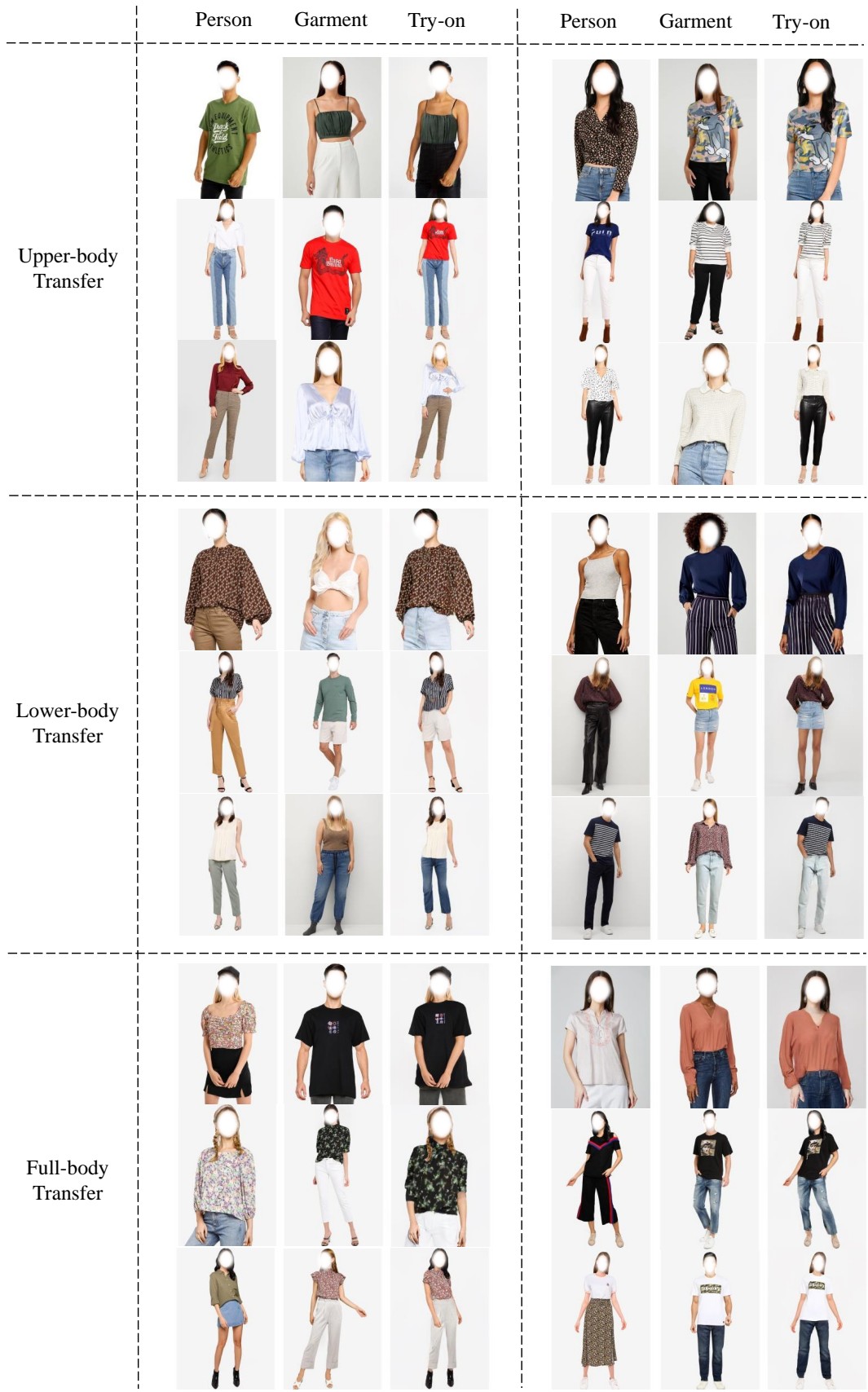

Figure 5: Virtual try-on results by our PASTA-GAN. Please zoom in for more details.

Figure 6: Virtual try-on results with high resolution ($512 \times 320$) by our PASTA-GAN. Please zoom in for more details.

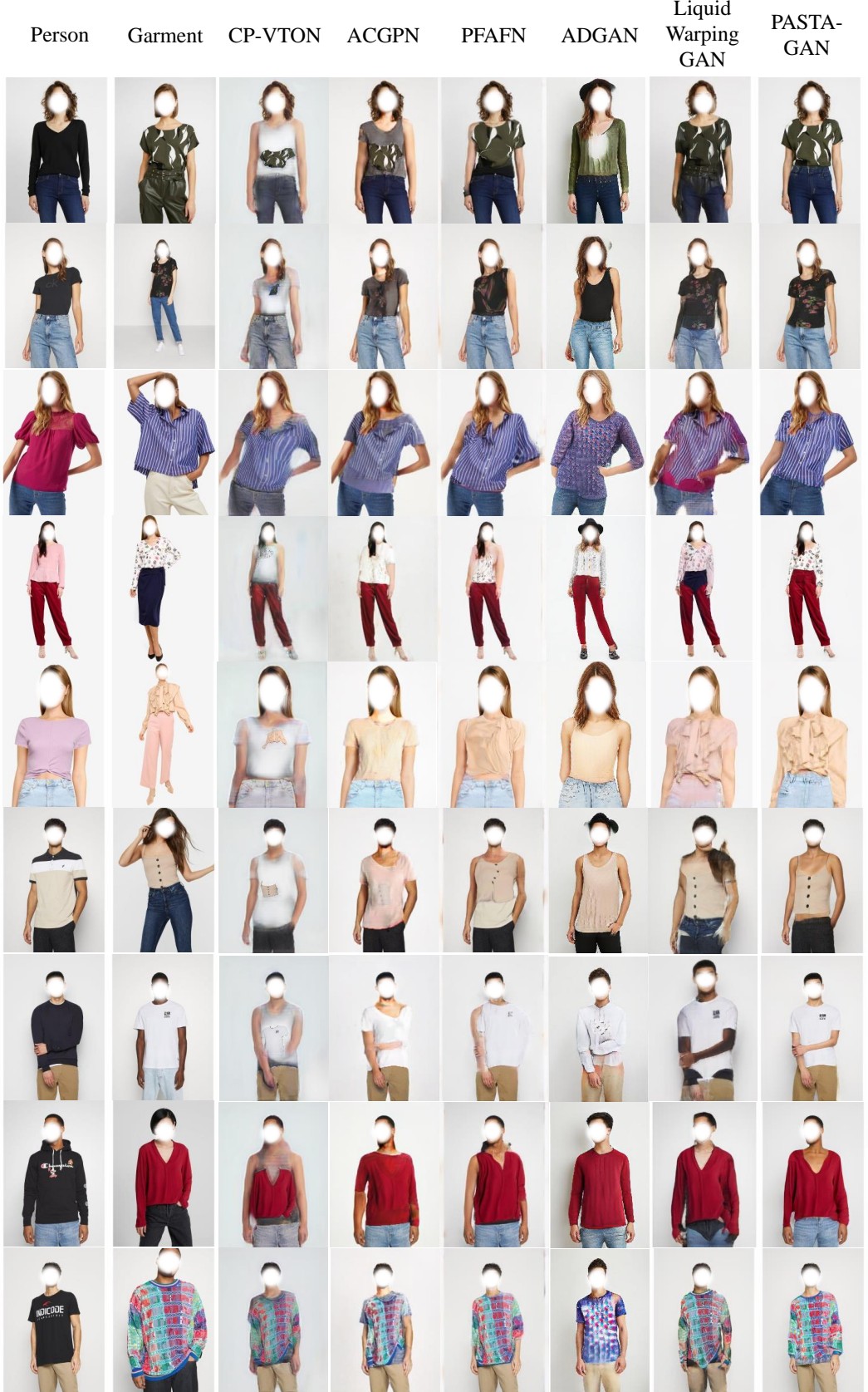

| Person | Garment | CP-VTON | ACGPN | PFAFN | ADGAN | Liquid Warping GAN | PASTA-GAN |
|--------|---------|---------|-------|-------|-------|---------------------|-----------|

Figure 7: Visual comparison among PASTA-GAN and the baseline methods under the unpaired setting on UPT dataset. Please zoom in for more details.

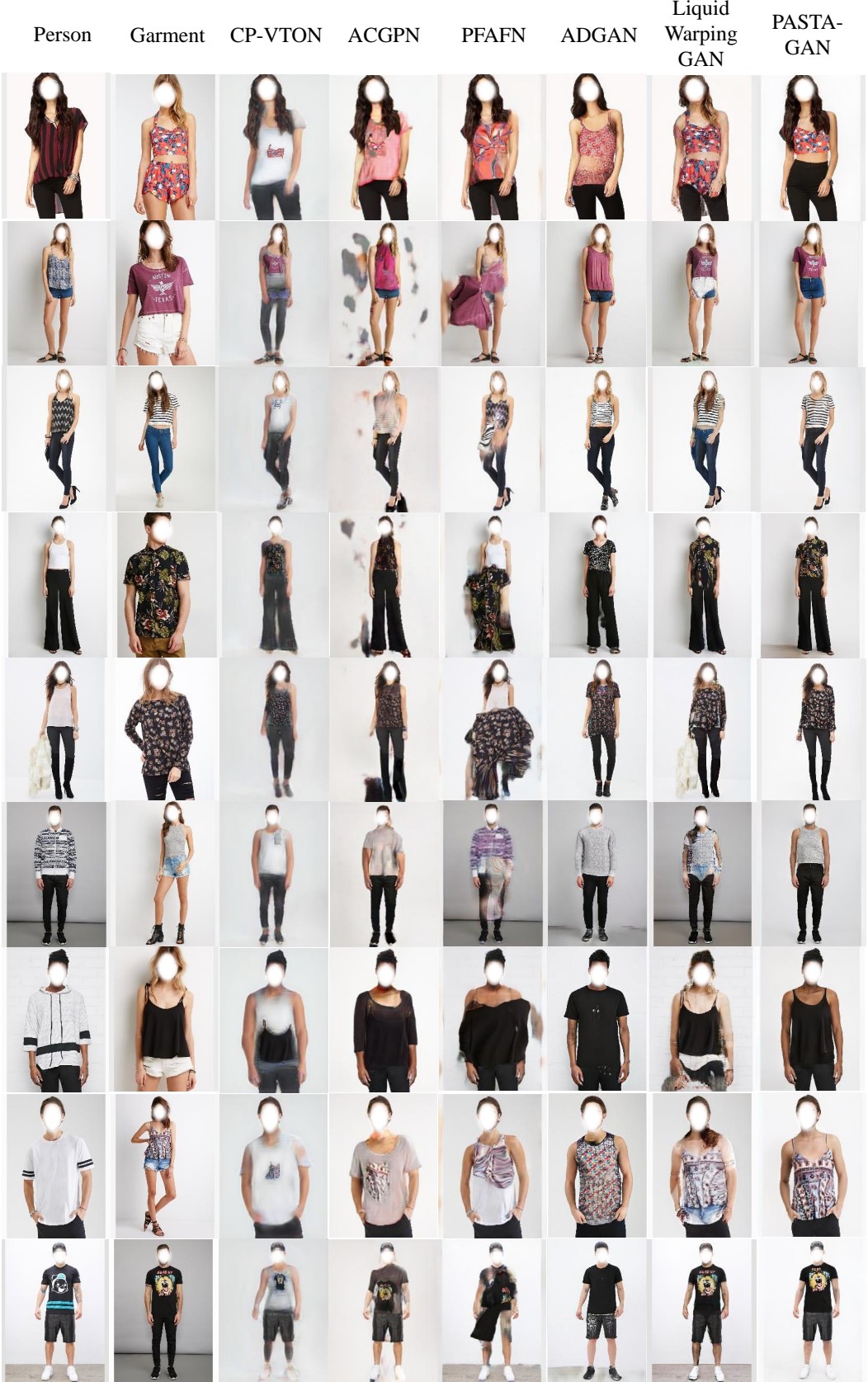

Figure 8: Visual comparison among PASTA-GAN and the baseline methods under the unpaired setting on DeepFashion dataset [9]. Please zoom in for more details.

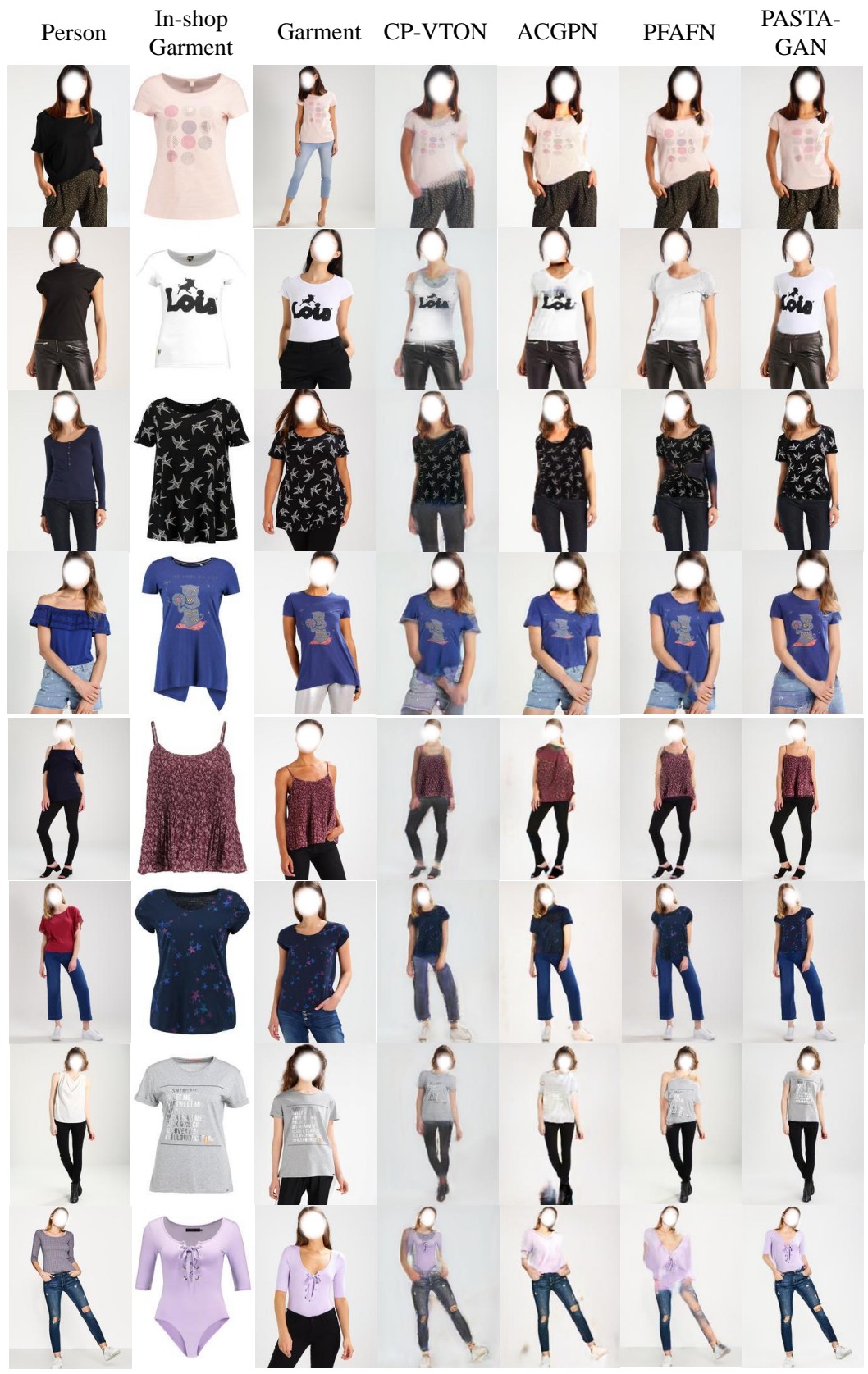

Figure 9: Visual comparison among PASTA-GAN and the baseline methods under their corresponding setting on MPV dataset [2]. Please zoom in for more details.

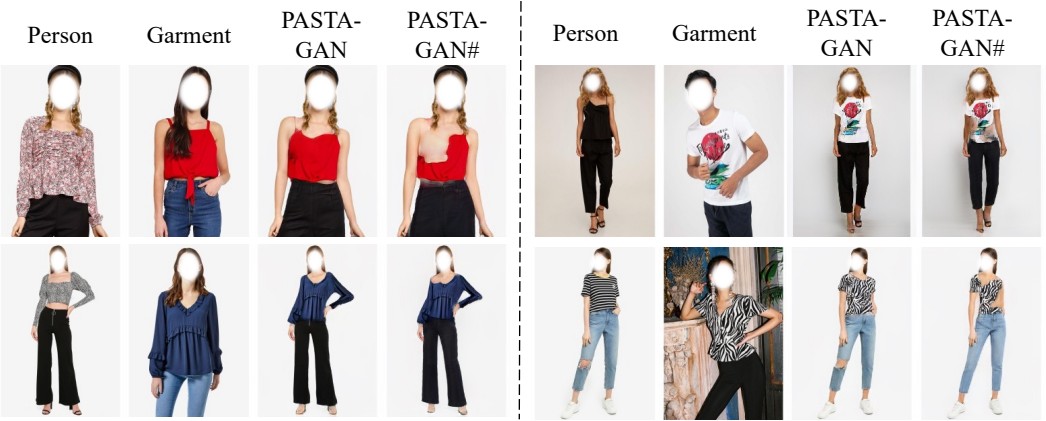

Figure 10: Qualitative results of the ablation study for the randomly erasing operation.