# OpenReview forum: "Towards Scalable Unpaired Virtual Try-On via Patch-Routed Spatially-Adaptive GAN"
_NeurIPS.cc/2021/Conference — NeurIPS 2021 Poster_

### Official Review · Reviewer_5Zj3 · 2021-07-14

**Rating:** 3
**Confidence:** 4

**Summary:**

This paper presents a human garment try-on pipeline. The key point of the proposed approach is to normalize each body part into a regular patch according to the body skeleton, and use this patch representation to extract the style and disentangle the spatial variation due to different body poses. The network is based on StyleGAN architecture, and adapted to accept style and pose conditions. Furthermore, to overcome the loss of the texture details, it also proposes to use the first-stage network to generate the clothed mask, and use a refining network to synthesize/inpaint the texture details based on the deformed patches aligned to the target skeleton.

**Limitations And Societal Impact:**

Most of the limitations are listed in the main review. Including unclear technical details and lack of explanation of the proposed network design.

**Main Review:**

It is not new to have a canonical representation for body appearance and garment to neutralize the pose variations, including using 3D UV space like DensePose or 2D patches like this paper. This paper does not provide enough insight why this simple patch extracting and normalization is better than previous solutions. Also, the normalization step is confusing, by mapping a rectangle (expanded vector between two joints) to a patch, a simple closed-form rotation+scaling should suffice, why there is a need to do least-squares optimization to get the matrix?

Regarding the network architecture, it looks pretty similar to the StylePoseGAN work (Style and Pose Control for Image Synthesis of Humans from a Single Monocular View). Although I guess it is still an unpublished work, why that method can produce better quality results without the need for the texture synthesis branch as the second stage? And in the texture branch, the feature inpainting method does not sound very convincing to me. What if the warped patches are not accurate, and how would it be to simply ask the network to generate the image with the warped patches as the soft condition input?

And my biggest concern is on the training losses. Why is it reasonable to have reconstruction loss between the source image and the result? They wear the same garment, but the body poses are not aligned.

---------------------

Addition to the final review:

I would like to thanks the authors for participating in the long discussion. While the discussion helped achieve a better understanding of most technical details, I am not fully convinced that this paper is ready for acceptance. My major concerns still exist:
1. The way how self-supervised training is done in the proposed method introduces a huge training-testing gap that is not negligible. The proposed augmentation approach does help improve robustness against occlusion but still suffers from unseen mismatches and other defects that are not seen during training. Please see the detailed comments.
2. The way how the paper handles patch cropping and normalization/deformation is presented as is. Many choices should be justified: Why optimization is needed for normalization? Why the proposed way to generate the cropping patch is the best? Why the patch does not cover the whole garment? But the paper does not provide enough experiments to convince the reader.
3. Lack of significant technical breakthrough. The proposed method is somehow similarly studied by other works on cloth try-on or human retargeting. It does have some interesting ideas, but I am concerned that how these ideas can really inspire follow-up works.

**Time Spent Reviewing:**

4

---

> ### Author Response · Authors · 2021-08-10
> **Response to Reviewer 5Zj3**
>
> Thank you for your detailed feedback. We believe that your main concern and low score are the results of a slight misunderstanding of the training mechanism of our proposed model. We will in the following try to clarify your concerns and will ensure that the relevant parts in the revised manuscript are rephrased to avoid such confusion. We sincerely hope you can reconsider and reevaluate our work. We address your main concern regarding the reconstruction loss first and then address the remaining questions following their original order:
>
> * **Training losses**: "Why is it reasonable to have reconstruction loss between the source image and the result?"
> * **Innovation and superiority of our canonical representation**: "The innovation of the canonical representation for body appearance and the superiority of the proposed 2D-pose-based strategy to the DensePose-based strategy."
> * **Reason for least-squares optimization**: "The reason for using the least-squares optimization to get the transformation matrix for patch deformation."
> * **Difference comparison with StylePoseGAN**: "PASTA-GAN is similar to StylePoseGAN."
> * **Need for the texture synthesis branch**: "Why StylePoseGAN can produce better quality results without the need for the texture synthesis branch as the second stage?"
> * **Refinement for coarse warped garment**: "What if the warped patches are not accurate, how would it be to simply ask the network to generate the image with the warped patches as the soft condition input?"
> * **Technical details**: "Unclear technical details and lack of explanation of the proposed network design."
>
>
> ## **Training losses**
>
> PASTA-GAN is designed to tackle the task of unpaired virtual try-on. This means that there are no image pairs available that can be used as ground truth. Therefore the model is trained in an unsupervised manner via image reconstruction, such that I_s and I_t are identical during training. During training, the source and the result image will therefore share both the same garment as well as the same pose, making it reasonable to use a reconstruction loss. Our proposed pose-guided patch-routed disentanglement module facilitates such a training procedure by decoupling the garment style from the shape information, avoiding trivial solutions where the model learns to only reconstruct the input image.
>
> We believe that this confusion is caused by the fact that Figure 2 illustrates the inference process, where I_s and I_t are different images. We will rephrase the caption to remove this source of misunderstanding.
>
>
> ## **Innovation and superiority of our canonical representation**
>
> Although canonical representations for body appearance and garments have been introduced by previous methods like StylePoseGAN[1], the strategy to obtain the canonical garment patches in our PASTA-GAN is completely different from the the previous DensePose-based strategy and has some clear advantages in the virtual try-on scenario.
>
> The garment patches obtained according to the DensePose[2] UV map can not guarantee the completeness of the source garment, since it can only extract the texture inside the region provided by the DensePose model and neglects the area outside of it. This can cause problems when considering loose clothing. Take as an example the long sleeve garments. If we utilize DensePose, the garment patch around the arm can only contain the garment texture inside the arm region, regardless of how loose the sleeve is. Such incomplete garment patches result in imprecise warped garments and will further impair the texture synthesis of the try-on result. In our PASTA-GAN, instead, we can account for this by adjusting the garment region through the patch size hyper-parameter.
> Besides, the DensePose is much more complicated than the 2D pose, which requires a more elaborate model for precise prediction and increases the computational cost of data pre-processing.
> We will explicitly explain the superiority of our 2D-pose-based strategy over the previous DensePose-based strategy for the virtual try-on scenario in the revision.
>
> Furthermore, previous approaches have used canonical garment patches for paired (supervised) human synthesis tasks (e.g. pose transfer), whereas we use them to enable the training of a model that can learn spatial-agnostic garment features in an unsupervised setting (see Line 37-43 and Line 60-64).
>
>
> ## **Reason for least-squares optimization**
>
> The garment patches obtained from the source garment can be quadrilaterals with arbitrary shapes (e.g., rectangle, square, trapezoid, etc.). The simple affine transformation (rotation+scale) can not tackle the transformation between an arbitrary quadrilateral and a square, while the perspective transformation can. Thus, given the four corresponding corner points of the source and normalized patch, the least-squares optimization is required to obtain the perspective matrix.
> We will clarify this in the revised version. We will also modify Figure 3 to consider the torso patch (which is a trapezoid patch) as an example, thereby illustrating that a perspective transformation is necessary for patch deformation.
>
>
> ## **Difference comparison with StylePoseGAN**
>
> While both PASTA-GAN and StylePoseGAN[1] inherit the architecture of the conditional StyleGAN2[3], there are several clear discrepancies between these two works.
>
> First, while StylePoseGAN considers the paired scenario and can train the synthesis network in a supervised manner, our PASTA-GAN is designed for the task of unpaired virtual try-on where no paired data is available. To enable training PASTA-GAN, therefore, addresses the challenging problem of training the synthesis network in an unsupervised manner.
> Second, instead of directly using StyleGAN2 as the architecture of the generator, we design a style synthesis branch and a texture synthesis branch with our proposed spatially-adaptive residual module in the generator, which separately aim to predict the precise garment mask and synthesize realistic try-on results with detailed texture.
> Third, our PASTA-GAN uses a completely different strategy to obtain the normalized garment patches. The advantage of this strategy over StylePoseGANs DensePose[2] approach has been described above.
> Finally, our PASTA-GAN further transforms the normalized patches to the target shape and obtains the warped garment, which is essential for the synthesis of the texture-preserved try-on results.
>
> Although StylePoseGAN is still an unpublished work, we will explicitly analyse the differences mentioned above in the revision.
>
>
> ## **Need for the texture synthesis branch**
>
> Compared to StylePoseGAN, which uses the paired data (i.e., the same person in different poses) for the supervised training, our PASTA-GAN is trained in an unsupervised manner as paired data is not available. This makes it more challenging for the network to learn the features of the garment texture.
> Further, the appearance details of the synthesized results from StylePoseGAN are dependent on the synthesis capability of the network, which is unable to completely retain the precise patterns and details of the source garment (e.g., logo, stamp, etc.). This issue has also been observed in previous work that build on StyleGAN (VOGUE, NHRR, etc.). PASTA-GAN proposes the spatially-adaptive residual module in the texture synthesis branch to introduce the warped garment features into the synthesis network, which are beneficial for complete preservation of the texture details.
>
> ## **Refinement for coarse warped garment**
>
> As illustrated in Line 200-215, our PASTA-GAN does not simply take the warped garment as the condition input of the synthesis network. Instead, it uses two separate operations to refine the coarse warped garment. Given the warped garment, PASTA-GAN first use the predicted garment mask from the style synthesis branch to remove the part of warped result outside the predicted mask. Then, PASTA-GAN uses the average garment feature to inpaint the misaligned region within the predicted mask. Thus, we argue that the refined and inpainted warped garment feature can guide the synthesis network to generate try-on results with texture details.
>
> ## **Technical details**
>
> We believe that we describe the technical details of our PASTA-GAN, including the motivation, specific technical process, and the detailed network architecture of the different sub-modules sufficiently. However, we are open to clarify any particular aspects in the revised version. In the following, we guide the reviewer to the relevant parts of the paper that address the above-mentioned points.
>
> The motivation of the patch-routed disentangling module, the two-branch StyleGAN-based synthesis network, and the spatially-adaptive residual module are provided in Line 114-129, Line 175-188, and Line 195-209, respectively.
> The specific technical process of the patch-routed disentangling is described in Line 130-148 while the specific process of the spatially-adaptive residual module is presented in Line 210-235.
> The details of the network design are illustrated in the first section of the supplementary, in which the network layers and operations are comprehensively depicted in Table 1 to Table 4.
>
> ---
>
> [1] Sarkar et al. "Style and pose control for image synthesis of humans from a single monocular view.", 2021.
>
> [2] Güler et al. "DensePose: Dense Human Pose Estimation In The Wild.", 2018.
>
> [3] Karras et al. "Analyzing and improving the image quality of stylegan.", 2020.

---

> > ### Comment · Reviewer_5Zj3 · 2021-08-12
> > **Response to the rebuttal**
> >
> > Thanks authors for the detailed response, which does provide helpful information for understanding the paper. Here I have some more questions and comments regarding the concerns in my original review.
> >
> > 1. Training losses
> >
> > I understand that self-reconstruction is performed during training to provide supervisions for reconstruction and other losses. However, my major concern is that self-reconstruction might not be solely sufficient for this particular task. There is no doubt that this self-supervision training strategy is pretty common in many image generation tasks. But the underlying assumption is that there is good disentanglement between the conditioning factors to the generator. Use style transfer as an example, assuming that we have ways to encode style S and content C from an image, and the target generator we want to train is G(S,C). We can definitely train G with I_A=G(S_A,C_A), and do inference with I_{AB}=G(S_A,C_B) to transfer the style from A to B, but this would only work if S and C are clean and complete representations of both style and content. Looking at this virtual try-on framework, we have pose and appearance as the factors, but the patch normalization step only handles very coarse piece-wise rigid pose of the human body, those non-rigid deformation, pose-dependent wrinkles, and tracking inaccuracy can still be seen in those normalized patches. With this shape information leaked to the appearance representation, the self-supervision becomes less effective when being generalized to new combinations of pose and appearance conditions. Furthermore, during training, the only thing the generator needs to learn is to approximate the perspective transformations to map the normalized patches back to their original shapes, which is much easier than dealing with the real testing cases that even perfect transformation would not produce convincing results, but the generator needs to also adjust the non-rigid deformation, align cross-patch discrepancies, and correct potential tracking errors or inconsistencies. Also, with self-supervised training, there will be much less misalignent between the warped mask and target mask (Fig.4). Therefore I am concerned if the proposed training strategy could handle this training-testing gap robustly. My question (or suggestion) is, would it be possible to add an unsupervised training branch that does I_{AB}=G(S_A,C_B) without reconstruction (because of no ground-truth) but with unsupervised losses like GAN, cyclye-consistency, etc. to help ensure the synthesizing quality on unpaired conditions? It would not fix all issues but I guess it could at least help. For example the GAN loss could be directly used on training pairs with pose and appearance from different references, that is what I was actually asking in the review.
> >
> >
> > 2. Innovation and superiority of our canonical representation
> >
> > I fully agree that densepose has issues on representing loose garment that falls outside the 3D body parts, and this is indeed the advantage of 2D image-space canonical mapping. However, there are some relevant works on doing 2D patch-wise cloth alignment with 3D representation for try-on that are very similar to this work. For example, in this work: Neural Rendering and Reenactment of Human Actor Videos, please see the pipeline Fig.2 that it renders cloth parts that align with the target skeleton and use a final generator that synthesizes the resulting image with these aligned patches as conditions. There are differences in how the parts are generated, but the overall idea is close. Please provide more insight on that and consider to discuss in future versions.
> >
> >
> > 3. Reason for least-squares optimization
> >
> > Thanks for the clarification. While I am still not 100 percent sure if I understand how the patches are generated with arbitrary shapes, I guess the parts with two joints will always produce rectangle shapes but the only exception is the torso part, am I right? If so I am still concerned that do this optimization for only one part is kind of ad-hoc if we agree that mapping between two rectangles does not need any optimization. Also, is the torso isosceles trapezoid? If so, then a linearly varying scaling along the axis would suffice. And even if the original shape is indeed arbitrary, we can always approximate a mapping using barycentric weights on the two triangles. If the proposed optimization is better, a comparison might be necessary.

---

> > > ### Author Response · Authors · 2021-08-16
> > > **Response to Reviewer 5Zj3**
> > >
> > > Thanks for your feedback and suggestion again. We argue there is still a slight misunderstanding of our PASTA-GAN. We will try to address all of your concerns in details and correct the misunderstanding by reiterating some technical details and pointing out the relevant parts in the original manuscript.
> > >
> > > In the following, we will address your concerns one by one:
> > > * **Training losses: information leak**: "With shape information leaked to the appearance representation, the self-supervision becomes less effective ..."
> > > * **Training losses: learning of PASTA-GAN**: "during training, the only thing the generator needs to learn is to approximate the perspective transformations to map the normalized patches back to their original shapes, ..."
> > > * **Training losses: training-testing gap of the misalignment region**: "With self-supervised training, there will be much less misalignment between the warped mask and target mask (Fig.4). ..."
> > > * **Training losses: unsupervised training branch**: "Would it be possible to add an unsupervised training branch that ..."
> > > * **Innovation and superiority of our canonical representation**
> > > * **Reason for least-squares optimization**
> > >
> > >  ### **Training losses: information leak**
> > > The concern of the reviewer is that the disentanglement between pose and appearance is inferior, and the information leak in the appearance representation will degrade the generalization of PASTA-GAN during testing. However, such an information leak is not an issue for our PASTA-GAN and does not influence the try-on result. Actually, our patch-routed disentanglement is not designed to decouple the human pose and the garment's appearance. As mentioned in Line 117-125, it aims to remove the inherent spatial information of the intact garment in the input person image, which will further promote the style encoder to extract spatial-agnostic garment features, encoding the information of the garment category and the overall color. The style encoder will not leverage the leaked information like pose-dependent wrinkles, since it is designed to extract spatial-agnostic garment features rather than the detailed appearance features. To synthesize realistic appearance detail in the try-on result, our PASTA-GAN introduces the spatially-adaptive residual module in the texture synthesis branch to leverage the warped garment features in the synthesis network.
> > >
> > > We guess the reason why the reviewer has such a concern is that he/she interprets the virtual try-on task as a style transfer based image synthesis problem. However, there exists an essential discrepancy between these two problems. While style transfer aims to synthesize a new image by integrating the content from one image and the style from another image, virtual try-on aims to transfer the garment onto the target person and preserve the garment details (e.g., shape, color, texture, etc.) as much as possible. Directly using the paradigm of style transfer for the virtual try-on task fails to retain the high-frequency information of the input garment, (i.e., texture, logo, etc.). For example, O-VITON[1] separately encodes the body shape and appearance information, and directly uses the appearance generator to take in the disentangled representations to synthesize the try-on results, which is similar to the traditional style transfer paradigm to some extend. Although the shape features and appearance features are clear and complete representations of both the body shape and the person's appearance, respectively, the synthesis results of O-VITON fail to preserve the high-frequency information of the input garments if no online optimization is performed.
> > >
> > > Thus, our PASTA-GAN does not inherit the style transfer paradigm. Instead, our PASTA-GAN incorporates the synthesis result and warped garment to guarantee the fidelity of the synthesis results and the preservation of the garment details. More specifically, our PASTA-GAN first proposes a patch-routed disentanglement module to separate the intact garment into normalized patches, which facilitates the style encoder to extract the spatial-agnostic garment features (garment category, color). Then in the style synthesis branch, our PASTA-GAN estimates the precise garment mask and synthesizes the coarse try-on result (without precise high-frequency garment information). In the texture synthesis branch, our PASTA-GAN uses the predicted garment mask to refine the coarse warped garment and uses the spatially-adaptive residual block to inject the warped garment feature into the synthesis network. Finally, our PASTA-GAN can synthesize the try-on results with high-frequency garment details.
> > >
> > > ### **Training losses: learning of PASTA-GAN**
> > > It seems that there is still a misunderstanding of our framework, which we hope to correct in the following.
> > >
> > > First, our generator is not designed to estimate the perspective transformation matrix. Actually, the patch warping procedure is learning-free. As mentioned in Line 156-164, given the target pose, we can directly calculate the perspective transformation matrix between the normalized patch and the target patch, without requiring any warping network. We regard such a learning-free warping approach as one of our contributions since most of the existing virtual try-on work requires paired training data to train an extra warping network.
> > >
> > > Second, during training, the style encoder takes the normalized garment patches as inputs and learns to extract the garment style feature. The style synthesis branch learns to synthesize the coarse try-on result and predict the garment mask, according to the extracted person identity feature and garment style feature. The texture synthesis branch learns to inpaint the feature of warped garment and synthesize the final try-on results, according to the extracted person identity feature, garment style feature, and the coarse warped garment. For the style encoder and the style synthesis branch, we argue that there is no difference for training and testing, since the data that these two modules can access is the same in both phases, namely, the normalized patches and the person pose information.
> > > For the texture synthesis branch, the stitched warped garment is more similar to the original intact garment during the training phase, since the normalized patches are deformed to their original shape. However, we do conduct some random erasing for the warped garment (Line 246-249) to imitate the real scenario in the testing phase to increase the generalization of our PASTA-GAN during the testing phase.
> > >
> > > ### **Training losses: training-testing gap of misalignment region**
> > > Actually, as mentioned in Line 246-249, during training we use two erasing mechanisms to randomly erase some regions of the warped garment to imitate the self-occlusion and misalignment in the warped garment in the testing phases. With such a randomly erasing mechanism, the training-testing gap can be largely reduced.
> > >
> > > ### **Training losses: unsupervised training branch**
> > > Thanks for your constructive suggestions. Adding an unsupervised training branch during training could be a promising direction to address the failure case of our PASTA-GAN (e.g., drastic pose change virtual try-on). However, we argue that the self-reconstruction training strategy used by PASTA-GAN is sufficient for the front-view setting of the unpaired virtual try-on. The experiment results illustrate that using such a training strategy, PASTA-GAN can outperform other baseline methods and obtain impressive try-on results.
> > >
> > >
> > > ### **Innovation and superiority of our canonical representation**
> > > We argue there are three main discrepancies between [2] and our PASTA-GAN, which make it incapable of tackling the unpaired virtual try-on task.
> > >
> > > First, in [2] the 3D character model is reconstructed from static posture images in different views. Such a 3D reconstruction procedure is completely contradictory to our unpaired setting since, for each person image, the corresponding images with other views are inaccessible under the unpaired setting.
> > >
> > > Second, during training, [2] uses monocular video as training data, which means it can access images of the same person with different poses. Thus there is abundant data for training the Character-to-Image translation network in a supervised manner, which largely reduces the difficulty of the training procedure.
> > >
> > > Third, [2] does not consider how to remove the inherent spatial information of each body patch, since it is trained in a supervised manner. Although it separates the body appearance into several patches, it does not normalize them into a canonical representation. For our PASTA-GAN, instead, the canonical representation is essential to extract the spatial-agnostic garment style features.
> > >
> > >
> > > ### **Reason for least-squares optimization**
> > > Thanks for your detailed feedback. As you mention in the comment, the patches obtained according to two joints are rectangles. The patch for the torso part can be a trapezoid or an arbitrary quadrilateral, which depends on the location of the joints for the left/right shoulder and the left/right hip. For convenience, we use the perspective transformation for the deformation of all kinds of garment patches (rectangle and arbitrary quadrilateral). Thus, we directly use the OpenCV API to calculate the perspective transformation matrix, which is implemented by using least-squares optimization. However, since the deformation method between the quadrilateral patches is not our contribution, we can simply replace the perspective transformation with other advanced approaches which have the same effect as the perspective transformation and are more time-efficient. We will try to compare the deformation effect and the running time of the perspective transformation and of the method that you mention in your comment.
> > >
> > > ---
> > >
> > > [1] Neuberger et al. "Image Based Virtual Try-on Network from Unpaired Data.", 2020.
> > >
> > > [2] Liu et al. "Neural Rendering and Reenactment of Human Actor Videos.", 2019.

---

> > > > ### Comment · Reviewer_5Zj3 · 2021-08-17
> > > > **Response to Authors**
> > > >
> > > > Thanks authors for the additional response. But unfortunately, I don't think it addresses most of my key concerns. And with all due respect, I disagree that I misunderstand the method. Below let me try to elaborate my opinions, if there is any misunderstanding please point it out.
> > > >
> > > > I would like to first clarify that I use style-content disentanglement merely as an example to discuss the effectiveness of self-supervision for multi-conditional generation, but not mean this virtual try-on problem falls into that paradigm, although they do share certain similarities. My major concern is not about the quality of the disentanglement, but the gap between the training and testing pairs, plus the sole use of self-reconstruction training strategy.
> > > >
> > > > If we look at the pipeline Figure 2(b), the input to the generation network backbone is 1) a set of normalized patches (P_n from I_s) sampled according to the source skeleton (J_s), and 2) the target skeleton (J_t). With self-reconstruction training, we have I_s==I_t and J_s==J_t during the training time. Therefore, even if we don't normalize the patches, but just crop I_s with the bounding box of the corresponding skeleton segment and paste it onto the white patch images without any transformation, the network can still learn to reconstruct I_s, right? Because no information is missing or conflicting, the network only needs to learn to simply put the patches back to the center positions according to J_t (high-level speaking). However, during inference, once J_s is no longer equal to J_t, the network won't be able to produce any reasonable results, because it assumes and utilizes this strong spatial correlation between P_n and J_t, even if we know it's incorrect. So, the key to make self-training work is to break this correlation and make P_n independent of any skeleton configuration. What this paper does is to normalize the patch with low-dimensional perspective transformation, to "remove the inherent spatial information" as the authors responded so that P_n can hopefully be independent of any skeleton, which means that as long as the training (P_n+J_s-->I_s) goes well, the inference (P_n+J_t-->I_st) will also work. While this does make sense, here comes my question: does this method really remove ALL the spatial information from the patches? I believe the answer is no. Take the arm as an example, with different elbow angles, both upper- and lower-arm not only do rigid or perspective transformations, but also involve complicated local non-rigid deformation of the silhouette and changes of inner cloth details, not to mention the inevitable error of the tracked skeleton. All spatial information not correctly removed by the normalization with remain in the patches. A simple way to understand is, if we warp P_n with J_s or J_t, we get totally different results: G_s is infinitely close to G_t with no cross-patch seams or shape misalignment, but G_t is broken into clear patches with jaggy silhouettes. This is the bad correlation I am concerned about. Using this kind of paired data for training, similar to unnormalized patches, the network will try to make the best use of the spatial correlation, such as assuming a simple transformation is enough to align all patches without seam, but the actual problem is much harder than the network assumes!
> > > >
> > > > The authors also claim that the design of this specific network architecture makes it less sensitive to this training-testing gap. But I cannot agree either. 1) I understand the style synthesis branch mainly focuses on synthesizing the mask of the garment, all texture details are synthesized during the texture branch. However, even if the style synthesis branch only does mask generation, the smoothness and correctness of the mask M_g also depend on the aforementioned spatial correlation between P_n and J_t. And I don't think some random data augmentation could bridge this gap easily. 2) For the texture branch, during training, G_t will be very close, if not identical, to G_s, if I understand correctly. Does this mean that the ground truth is fed to the network during training? 3) One may also argue that thanks to the feature representations, the style encoder tends to only encode globally consistent information for structure and appearance instead of spatially varying local details. But I don't think there is any theoretic guarantee, and given that the style synthesis branch needs to synthesize detailed mask shapes, the feature code w has to contain spatial information where incorrect correlation has the chance to leak.
> > > >
> > > > Regarding other issues, such as relevant papers and normalization optimization, I think I have reached a confident understanding. Thanks!

---

> > > > > ### Author Response · Authors · 2021-08-21
> > > > > **Response to Reviewer 5Zj3**
> > > > >
> > > > > Thanks for your response. We really appreciate the clarifications of your concerns. The reviewer's major concern is that, during training, the normalized patches still contain some spatial information of the source person image, which results in the "bad correlation" between the normalized patches and the source pose, and makes it difficult to generalize to the inference phase.
> > > > >
> > > > > We believe that the reasons for the reviewer's concern are due to the following two reasons. First, we and the reviewer have a slightly different understanding of the specific phrase "spatial information". Second, the reviewer argues that, to train our PASTA-GAN, all spatial-related information should be completely removed from the normalized patches, while we believe that normalized patches with the essential spatial information removed are sufficient for the self-reconstruction training of our PASTA-GAN. Both of these will be clarified below.
> > > > >
> > > > > In the following, we will try to address all of the reviewer's concerns by:
> > > > > * **Spatial information**: Clarifying the exact definition of the spatial information in our paper and explaining why our spatial-agnostic normalized patches are sufficient for the training of our PASTA-GAN;
> > > > > * **Information leak**: Analysing what influence the information leak problem may bring to our style synthesis branch and texture synthesis branch, and how our PASTA-GAN removes the potential negative influence;
> > > > > * **Technical details**: Reiterating some technical details to clarify some implementation details.
> > > > >
> > > > > ### **Spatial information**
> > > > > Let us first clarify the exact technical meaning of the "spatial information" defined in our paper. Spatial information in this context refers to the combination of the following three parts: the location, the orientation, and the relative size of the garment patch in the person image. More specifically, the first two parts are influenced by the human pose while the third part is determined by the relative camera distance to the person.
> > > > > After conducting the normalization process, for a particular type of garment patch (e.g., patch around the left lower arm), patches from the different person images will be normalized to a square patch with the same orientation (e.g., horizontal), regardless of their original location, orientation, and size. Thus, we argue that our patch-routed disentanglement module can remove the essential spatial information which is detrimental to the self-reconstruction training procedure for garment mask estimation.
> > > > >
> > > > > Taking as inputs the normalized garment patches, our style encoder and style synthesis branch can not directly leverage this spatial information in the person image to estimate the target garment mask and synthesize the coarse try-on result, instead, they learn to exploit the pose information from the input pose and the garment style information (i.e., garment category and overall color) from the input normalized patches. Actually, our normalized patches naturally encode garment category information. For example, for the long sleeve, all of the upper patches contain parts of the garment, while for the sling, the patches near the arm region do not contain any garment part. Thus, it is straightforward for the style encoder and style synthesis branch to capture the garment category information by judging whether there is a garment part inside a particular type of patch or not.
> > > > >
> > > > > ### **Information leak**
> > > > > The reviewer's concern seems to be that the non-rigid transformation in the arm region (caused by some particular elbow angles) may lead to an information leak of the pose, which impair the mask estimation in the style synthesis branch and warped garment in the texture synthesis branch during the testing phase.
> > > > > Before addressing this concern, we argue that it is necessary to clarify the usages of the normalized garment patches in our PASTA, which will help us analyse the potential impacts of such an information leak for different modules and how our PASTA-GAN addresses them.
> > > > >
> > > > > Our normalized garment patches have two effects. For the style encoder and the style synthesis branch, the normalized garment patches provide the garment style information, which is essential for the estimation of the garment mask. For the texture synthesis branch, the normalized garment patches are warped to the target shape and stitched together to form the coarse warped garment.
> > > > >
> > > > > For the style synthesis branch, we argue that the information leak that the reviewer is concerned about will be ignored by the network.
> > > > > Since we separate the garment around the arm region into two patches (i.e., upper- and lower-arm patches), for the particular type of patch (e.g., left lower arm patch), the shape of the garment in the normalized patches from different poses can be regarded as pose-agnostic. The reviewer may still argue that, with some particular elbow angles, there will be some bending or wrinkle silhouettes at the tail of the patches. However, we believe that such obscure information is not enough to provide any clear information about the pose. Furthermore, compared to estimating the garment mask by using such obscure information in the spatial-agnostic normalized patch, the style encoder and the style synthesis branch tend to exploit the pose information from the pose input and the category information from the normalized patches, since they are much more straightforward and intuitive.
> > > > > Our experiment results displayed in the paper can further verify that the input pose and the normalized patches can provide sufficient information about the pose and garment category for the style synthesis branch to estimate the precise garment mask.
> > > > >
> > > > > For the texture synthesis branch, we admit that, for most cases, due to the difference between the source and target poses, warping the normalized patches according to the source pose can obtain more precise warped results when compared to warping the normalized patches according to the target pose. However, note that the pose differences are not the sole reason for cross-patch seams or shape misalignment in the warped result.
> > > > > Since the garment patches are cropped from the person's image according to the pose keypoints, there is no guarantee that the garment patches cover the whole garment during training. Please refer to Fig 3 in the paper. The patch for the torso region does not cover the garment around the torso region. When warping these kinds of normalized patches back to the original shape, the stitched warped results can be different from the real garment. Thus, also during training, the cross-patch seams or shape misalignment can exist in the warped results for some cases.
> > > > > Nonetheless, during training, in the texture synthesis branch, we leverage erosion operations on the normalized patches and random erasing on the warped results to further imitate the cross-patch seams and shape misalignment for the warped result in the testing.
> > > > >
> > > > > ### **Technical details**
> > > > >
> > > > > * Mechanism of our patch warping
> > > > >
> > > > > To eliminate potential confusion, we would like to clarify that, the transformation between the two corresponding patches (i.e., source-normalized patches or normalized-target patches) is learning-free and our network does not need to assume any transformation for patch warping.
> > > > >
> > > > > * Data augmentation for the normalized patches
> > > > >
> > > > > What we want to explain is that, for the style synthesis branch, the normalized patches are directly sent to the network without any data augmentation. Instead, in the texture synthesis branch, for the warped garment, we conduct the erosion operation on the normalized patches and conduct the random erasing operation on the warped result.
> > > > >
> > > > > * The similarity of the warped garment and the ground truth
> > > > >
> > > > > As we mention in Line 246-249 in the paper and answered in last response comment, during training, we conduct random erasing on the warped garment to reduce the training-testing gap.
> > > > >
> > > > > * Explanation about the feature from the style encoder
> > > > >
> > > > > Though there is not any theoretic guarantee that the style encoder can not encode the spatially varying local details, we have tried to explain it intuitively in Line 178-181. First, the style encoder projects the normalized patches into a one-dimensional vector, resulting in loss of high frequency information. Second, due to the large variety of garment texture, learning the local distribution of the particular clothing details is highly challenging for the basic synthesis network.
> > > > >
> > > > > Besides, similar opinions have been fronted and have been verified in [1].
> > > > >
> > > > > * The spatial information in feature w
> > > > >
> > > > > Actually, the style feature w can only embed the garment style feature, since it is derived from the spatial-agnostic normalized patched. As for the spatial information, it is embed in the person identity feature, since it is derived from the the input pose.
> > > > >
> > > > > ---
> > > > >
> > > > > [1] Neuberger et al. "Image Based Virtual Try-on Network from Unpaired Data.", 2020.

---

> > > > > > ### Comment · Reviewer_BGd4 · 2021-08-31
> > > > > > **Garment patches Gt during training**
> > > > > >
> > > > > > By reading above discussions with reviewer 5Zj3, I'm curious how the patches Gt look like during training as the source and target poses are the same for self-supervised learning. The transformation $H_{n\arrow t}$ is just the inverse of $H_{s\arrow n}$, or there is no need to calculate the transformation.
> > > > > >
> > > > > > The data augmentation in Line 245-249 seems very important for reducing the gap between training and test. I wonder if there is any ablation study with/without this step.

---

> > > > > > > ### Author Response · Authors · 2021-09-01
> > > > > > > **Response to Reviewer BGd4**
> > > > > > >
> > > > > > > Thank you for your reply and for giving us the chance to address your concerns. We will in the following respond to your first question regarding the patches during training and provide visual examples. We have also started on the requested ablation study and will provide the results in the next 24 hours.
> > > > > > >
> > > > > > > 1. **How the patches Gt look like during training?**
> > > > > > >
> > > > > > > In our implementation, during training, although the source and target poses are the same, the coarse warped garment Gt is **not** identical to the intact source garment Gs. The reasons for this are three fold: (1) the quadrilateral crop for Gs is by design not seamless/perfect, there will accordingly often exist some small seams between adjacent patches in Gt, and also incompleteness along the boundary of the torso region; (2) in the normalized patches Pn, the individual garment patches belonging to arms or legs are randomly dropped out with a fixed probability; (3) after the Gt is obtained by back-projecting the Pn, it is further erased by applying the random masks proposed in [1] to imitate the self-occlusion of the source person.
> > > > > > > Thus, due to the above-mentioned three operations, the warped garment Gt that is sent into the network is quite different from the intact garment Gs cropped from the source person image during training.
> > > > > > >
> > > > > > > In the following anonymous link [Warped_garment](https://figshare.com/s/d9750bf34c977c9283f4), we illustrate the intact source garment Gs, the warped garment directly by stitching warped patches Gt', and the warped garment after the random erasing process Gt.
> > > > > > >
> > > > > > > 2. **Can H_n2t be the inverse of H_s2n during training?**
> > > > > > >
> > > > > > > Yes, we agree that calculating the inverse of H_s2n would be an alternative implementation to obtain H_n2t.
> > > > > > > We further conduct an experiment to compare the computational cost between the two different implementation to obtain H_n2t and notice a negligible difference in computational cost (ours: 2.0266e-5 seconds v.s. inverse: 6.6638e-5 seconds). Note, "ours" is computed using OpenCV's perspective transformation matrix implementation, while the "inverse" is computed using numpy.
> > > > > > >
> > > > > > > ---
> > > > > > >
> > > > > > > [1] Liu et al. "Image inpainting for irregular holes using partial convolutions.", 2018.

---

> > > > > > > > ### Author Response · Authors · 2021-09-02
> > > > > > > > **Results of the required ablation study**
> > > > > > > >
> > > > > > > > First of all, we would like to thank the reviewer for suggesting this ablation study, as it also nicely addresses the concerns of Reviewer 5Zj3.
> > > > > > > > In the following, we provide results for the requested ablation study.
> > > > > > > >
> > > > > > > > We train our PASTA-GAN without the random removal process described in Line 246-249 and denote it as PASTA-GAN#. Then we compare PASTA-GAN# with the full PASTA-GAN both quantitatively and qualitatively.
> > > > > > > > For the quantitative result, the FID score of PASTA-GAN# increases from 7.851 to 12.531 (lower is better) compared to the full PASTA-GAN model.
> > > > > > > > The qualitative results further illustrate that the random removal process is essential to reduce the train-test gap and enhance the generalization of our proposed PASTA-GAN. PASTA-GAN# tends to fail at synthesizing precise texture in regions which are occluded by a body part in the source person image(e.g., hair, arms, etc.). We provide some visual comparisons in the following anonymous link [ablation results](https://figshare.com/s/bdebabba32cfb83cf6b9).
> > > > > > > >
> > > > > > > > Also, please note that although the results of PASTA-GAN# are less satisfactory, the model still succeeds in fitting the source garment to the target person pose J_t. In other words, even without random erasing, our network will not be affected by the information leakage, which is the main concern of Reviewer 5Zj3, but instead still learns to reconstruct the source garment following the guidance of the target pose J_t. Therefore, we argue that the information leakage of J_s is negligible compared to the information of J_t during inference.
> > > > > > > >
> > > > > > > > We will include this discussion in the revised manuscript.

---

> > > > > > > > > ### Comment · Reviewer_5Zj3 · 2021-09-06
> > > > > > > > > **Comments to the new experiments**
> > > > > > > > >
> > > > > > > > > Thanks authors for the new results, they do provide additional information to help understand and evaluate the method. I have some last-minute comments on the new results:
> > > > > > > > > 1. The authors claim that during training G_s will not be equal to G_t even with the same skeleton. But I believe it's somehow due to the implementation of the transformation. I can hardly imagine that warping a rectangle (the arm for example) using identical keypoints can result in such a large deviation as shown in "Warped_garment". This could not be claimed as the "benefit" of the proposed method just because the deformation quality is not good.
> > > > > > > > > 2. I agree randomly dropping parts of the mask could improve the robustness of training. But what concerns me is the training-testing gap of the patch alignment, which is in another dimension to occlusion that might be handled with random dropping.
> > > > > > > > > 3. By looking at the results, I realize another potential issue of the method. The patches are always smaller than the actual garment in most cases, especially the torso part that the sides of the waist are never covered by the torso patch, then how could the network be able to synthesis the whole garment if some part is never seen?
> > > > > > > > > 4. I agree that the additional ablation does prove the value of the augmentation, but as I described above, it is parallel to my concern.

---

> > > > > > > > > > ### Author Response · Authors · 2021-09-07
> > > > > > > > > > **Response to Reviewer 5Zj3**
> > > > > > > > > >
> > > > > > > > > > The reviewer argues that the difference between Gs and Gt is due to the poor quality of the patch transformation.
> > > > > > > > > > However, we believe that there exists a slight misunderstanding of some of the warped results that we provided in the second response to Reviewer BGd4[Warped_garment](https://figshare.com/s/d9750bf34c977c9283f4). For the warped result in the 3rd row, the large deviation for the right forearm patch between Gs and Gt is due to the patch cropping and is not related to the learning of the patch transformation. Patches are cropped using keypoints predicted by OpenPose and if the patches do not cover the whole garment **and/or** do not perfectly align with adjacent patches, there will be a deviation (even with identical keypoints). In this case, the root cause is due to the underperformance of OpenPose when faced with side views.
> > > > > > > > > >
> > > > > > > > > > To make this more clear, we would like to provide the process of patch-routed deformation for these examples in the following anonymous link [patch routed deformation](https://figshare.com/s/707b91341acfe8d57e89). The above mentioned case is illustrated in row 3. Considering the first two rows, with an accurate OpenPose keypoints estimation, the cropped patches successfully cover most of the garment in the source image and the large deviation does not occur between Gs and Gt. Here Gs* illustrates the source patches overlaying Gs and Gt'* illustrates the target patches overlaying Gt' (Gt before applying the erasing procedure).
> > > > > > > > > > However, as explained in the previous responses (21st, August@Reviewer 5Zj3 and 1st, September@Reviewer BGd4), there usually exist some small seams between the adjacent patches in Gt (see figure above), which makes it different from Gs, and we also apply random dropout/erasing on the warped results to further enlarge the discrepancy between Gs and Gt.
> > > > > > > > > > Moreover, while we do not claim that the large deviation is solely a benefit, it **does** regularise the model, which is important in unsupervised reconstruction tasks. Our model is further **robust** to these small deviations and we will elaborate this in the next paragraph.
> > > > > > > > > >
> > > > > > > > > > The reviewer's second concern is the training-testing gap of the patch alignment. We argue that it is challenging to completely reduce the training-testing gap (especially the patch alignment mentioned by Reviewer 5Zj3) since our PASTA-GAN is trained in a completely unsupervised manner. Nevertheless, we do apply operations to reduce the training-testing gap of the patch alignment and design a particular module to address the patch misalignment during testing. Before we explain the specific operation and the proposed module, let us first analyze what kind of misalignment our PASTA-GAN may encounter during testing. As we illustrated in Figure 4 in the paper, there are two types of misalignment during testing, namely, the misalignment inside and outside the target garment shape. For the inside misaligned regions (i.e., the orange region in Figure 4), during testing, our PASTA-GAN uses the spatially-adaptive residual module to synthesize the appropriate texture according to the warped garment features. During training, to imitate such inside misalignment, we conduct random erasing on the warped garment, and the network is forced to learn how to inpaint the inside misaligned region. For the outside misaligned regions (i.e., the green region in Figure 4) our PASTA-GAN applies the garment mask predicted by the style synthesized branch to directly exclude such outside misaligned regions in the warped garment during testing. Note that, since the style synthesis branch does not take as input the coarse warped results, the outside misalignment of the coarse warped garment will not influence the accuracy of the predicted garment mask and the predicted mask is precise enough to refine the coarse warped garment.
> > > > > > > > > > The impressive qualitative results in our paper further verify that our PASTA-GAN can handle the patch misalignment and generate realistic try-on results.
> > > > > > > > > >
> > > > > > > > > > The reviewer's third concern is that sometimes not all the garment area is covered by the patches and how the network can then synthesize these missing areas.
> > > > > > > > > > We argue that this issue is the same as the inside misalignment problem mentioned above, and it is therefore addressed by introducing the spatially-adaptive residual module into the PASTA-GAN and conducting random erasing on the warped garment during training to force the network to learn how to synthesize proper texture in the lost part. The left side example in the third row of Figure 5 can be a pretty intuitive example, where PASTA-GAN is generating realistic and complex texture around the waist region, which the torso patch is not completely covering.
> > > > > > > > > >
> > > > > > > > > > We hope that we addressed the reviewers' questions in a comprehensive manner, and are more than happy to provide any further clarification.

---

> ### Author Response · Authors · 2021-09-10
> **Response to the Updated Review of Reviewer 5Zj3**
>
> As the discussion phase is coming to its end, we would like to provide another clarification and reply to the updated review of Reviewer 5Zj3. In particular:
>
> 1. Our approach, does not neglect the training test gap. We design a purposeful designed module in the network that reduces this gap and in addition make use of task-inspired data augmentation. This allows us to reduce the gap sufficiently as illustrated by all our qualitative and quantitative results that show PASTA-GANs ability to generalize to new data. Given the unsupervised nature of the model, a reconstruction approach is in our view the only sensible approach. More details can be found in our previous response (7th, September@Reviewer 5Zj3).
>
> 2. As stated in our previous responses (10th, August@Reviewer 5Zj3 and 16th, August@Reviewer 5Zj3), the optimization is not used for the patch normalization, instead, it is used for the calculation of the perspective transformation matrix. This transformation is chosen based on its simplicity and efficiency. While alternative transformations could be chosen, the novelty of our approach does not lie in the exact choice of the transformation but in utilizing the patches to enable unsupervised training of the model. The proposed patch approach achieves this in an efficient manner (see experimental evaluation). Regarding alternative patch-extraction approaches, such as densepose, we have provided our reasoning in the response (10th, August@Reviewer 5Zj3). Further, in (21st, August@Reviewer 5Zj3 and 7th, September@Reviewer 5Zj3) we additionally provide more detailed descriptions and visualizations of the patching process, which explain why the patches do not cover the whole garment. We will include this material and arguments in the revised version.
>
> 3. Regarding the novelty, this is the first attempt to address the unpaired virtual try-on task within an end-to-end and scalable framework. It achieves impressive results, while not requiring additional auxiliary data or extensive online optimization. Note, while other approaches for **supervised** cloth try-on or human retargeting have used patch-based representations, they did not use them to **facilitate unsupervised** training, which in this work is the main reason for introducing them. Thus, we argue that there exists a sufficient amount of novelty and that this work will inspiring additional unsupervised follow-up works.

---

### Official Review · Reviewer_4awU · 2021-07-15

**Rating:** 8
**Confidence:** 4

**Summary:**

The paper creates realistic virtual try-on results by modifying the StyleGAN2 architecture and using the garment appearance as the style modulation. In particular, to preserve the details of the target garment, the garment patches are warped and used as reference for the output image. The proposed method significantly outperforms the existing baselines thanks to the photorealism brought by preserving the input patches. The improved quality is backed by FID, user study, and analysis via ablations.

**Limitations And Societal Impact:**

The authors discussed both positive and negative implications of the work in Conclusion.

**Main Review:**

I am generally satisfied with the clarity of writing, several insights and conceptual contributions, practicality of the application, and evaluation.

While the synthesis models based on neural networks produce nice results, it is very challenging to use it for retaining the precise details and patterns of the input image in the process of image manipulation, such as logos on t-shirts. For example, many GAN inversion works (Image2StyleGAN Abdal et al, 2020, StyleGAN encoder Richardson et al., CVPR 2021) can't embed complex patterns on hats (see Figure 11 of https://arxiv.org/pdf/2008.00951.pdf). The proposed method nicely handles this by first warping the garment to retain the details, and then using deep networks to refine the final output. I think this combination of non-parametric sampling and parametric approach will inspire future works in image manipulation using deep networks.

The writing is clear, and the authors comprehensively listed the details of the method, including the precise network architecture, insights (L175-181), details about implementing baselines (L271-281), failure cases and ethical concern.

The evaluation clearly supports the improved quality of the proposed method, both qualitatively and quantitatively. The FID is significantly better, likely because the input patches are recycled in the output.

Having said that, here are some weaknesses and suggestions.

Resolution is 256px. Can it be higher?

The quality of the final output depends on the quality of the warped garment. Therefore, when the pose change is too drastic to find a nice warp, the quality of the output will degrade as in Figure 3 of Supp Mat. Likely because of this reason, most results of the submitted paper are frontal-facing pose. Still, I think this limitation is not important, as it is reasonable to assume that the target garment will contain at least one frontal facing picture in the virtual try-on task.

The human evaluation protocol could be performed at larger scale, because 30 volunteers may not represent unbiased population.

The labeling of ablated models was too difficult to follow. Could you use more intuitive names? Such as “-TSB, intact-garment”.

It dismisses a line of work [19, 27, 26, 25] because their methods require images of the same person in different poses. However, this may not be a harsh requirement on the fashion dataset, as these types of images can be readily found on the Internet. In particular, the authors already evaluate their method on the DeepFasion dataset, and StylePoseGAN[25] shows nice results on this dataset. Ideally, some side-by-side comparison to the SOTA methods such as [25] would be nice, while clearly stating that the training requires different amount of supervision. This is merely a suggestion for future revisions, since [25] is still only on arxiv.

In particular, it does share some similarity to StylePoseGAN[25]. Both inherit StyleGAN2 and modify it to a conditional version. The garment style is used as style modulation. The garment style information is normalized to some canonical pose in both works. Considering StylePoseGAN is a recent work and still only on Arxiv, similarity to StylePoseGAN does not affect my rating, but it would be nice to further discuss similarities and differences, because the authors already cited the work.


**Time Spent Reviewing:**

4

---

> ### Author Response · Authors · 2021-08-09
> **Response to Reviewer 4awU**
>
> We are grateful for your comprehensive and encouraging feedback. We are pleased that you appreciate the technical innovation, the convincing experiment analysis, the thorough description, and the writing clarity. In the following, we will respond to your constructive suggestions and questions:
>
> * **Higher resolution results**: "Resolution is 256px. Can it be higher?"
> * **Degradation for drastic pose change**: "The quality of the output will degrade when the pose change is too drastic."
> * **Human evaluation protocol**: "The human evaluation protocol could be performed at larger scale, because 30 volunteers may not represent unbiased population."
> * **Ablation labeling**: "The labeling of ablated models was too difficult to follow. Could you use more intuitive names? Such as “-TSB, intact-garment”.
> * **Comparison with StylePoseGAN**
> * **Similarity and difference comparison with StylePoseGAN**
>
> ## **Higher resolution results**
>
> While we only present results with a resolution of 256x256 in the paper, this is not a limitation of the PASTA-GAN, which can synthesize higher resolution results. However, as the main focus of this work is on scaleable unpaired virtual try-on, we adapted the same resolution as our main baselines (i.e. ADGAN[1], Liquid Warping GAN[2]) for fair comparisons.
>
> We will display additional try-on results with a resolution of 512x512 in the revision.
>
> ## **Degradation for drastic pose change**
>
> Indeed, our PASTA-GAN mainly focuses on the unpaired front view virtual try-on and will obtain inferior performance when facing complicated and scare poses. We agree with you that this will most likely not be a big limitation in practice. However, it is an interesting problem to tackle unpaired virtual try-on with drastic pose change and could potentially be approached by incorporating additional 3D information. We will explore this more thoroughly in future work.
>
> ## **Human evaluation protocol**
>
> We will invite more volunteers to participate in our questionnaires to further improve the human evaluation quality.
> However, given the significant improvements in the human evaluation scores across all datasets on a somewhat subjective task of choosing the most realistic synthesized image, we believe that results will be similar.
>
> ## **Ablation labeling**
>
> Thanks for your constructive suggestion. We used these abbreviations due to space constraints but understand that this might have impacted clarity. In the revised version, we will leverage the additional content page and use more intuitive names.
>
> ## **Comparison with StylePoseGAN**
>
> Thank you for the suggestion. We will add StylePoseGAN[3] as one of our baselines if their code is released in time. We agree that such a comparison will further strengthen our proposed method. In any case, we will add a discussion of the similarity and differences (see next paragraph).
>
> ## **Similarity and difference comparison with StylePoseGAN**
>
> We will add a discussion of the similarities and differences between PASTA-GAN and StylePoseGAN[3].
>
> As for the similarity, as mentioned in your comment, both methods are based on conditional StyleGAN2[4], conditioned on the pose, and use garment feature as the input of the modulation layer. Besides, both methods use a canonical garment representation.
>
> However, there are four main differences between PASTA-GAN and StylePoseGAN that can be summarized as follows.
> First, while StylePoseGAN considers the paired scenario and can train the synthesis network in a supervised manner, our PASTA-GAN is designed for the task of unpaired virtual try-on where no paired data is available. Therefore, to enable training, PASTA-GAN addresses the challenging problem of training the synthesis network in an unsupervised manner.
> Second, instead of directly using StyleGAN2 as the architecture of the generator, we design a style synthesis branch and a texture synthesis branch with our proposed spatially-adaptive residual module in the generator, which separately aim to predict the precise garment mask and synthesize realistic try-on results with detailed texture.
> Third, our PASTA-GAN obtains the garment patches according to the 2D pose, while StylePoseGAN obtains the garment patches according to the DensePose[5] UV map. Note that the garment patches obtained according to the DensePose UV map can not guarantee the completeness of the source garment, since it can only extract the texture inside the region provided by the DensePose model and neglects the area outside of it. This can cause problems when considering loose clothing. Take as an example the long sleeve garments. If we utilize DensePose, the garment patch around the arm can only contain the garment texture inside the arm region, regardless of how loose the sleeve is. Such incomplete garment patches result in imprecise warped garments and will further impair the texture synthesis of the try-on result. In our PASTA-GAN, instead, we can account for this by adjusting the garment region through the patch size hyper-parameter.
> Finally, our PASTA-GAN further transforms the normalized patches to the target shape and obtains the warped garment, which is essential for the synthesis of the texture-preserved try-on results.
>
> ---
>
> [1] Men et al. "Controllable person image synthesis with attribute-decomposed gan.", 2020.
>
> [2] Liu et al. "Liquid warping gan: A unified framework for human motion imitation, appearance transfer and novel view synthesis.", 2019.
>
> [3] Sarkar et al. "Style and pose control for image synthesis of humans from a single monocular view.", 2021.
>
> [4] Karras et al. "Analyzing and improving the image quality of stylegan.", 2020.
>
> [5] Güler et al. "DensePose: Dense Human Pose Estimation In The Wild.", 2018.

---

### Official Review · Reviewer_vykN · 2021-07-17

**Rating:** 6
**Confidence:** 4

**Summary:**

This paper focuses on virtual try-on based on unpaired images. The authors propose to disentangle the style and spatial information of each garment, and generate it under the condition of homography transformed normalized patches. The authors also introduce a new unpaired dataset. Compared with the paired and unpaired state-of-the-art methods, both quantitative and qualitative results prove the effectiveness of the proposed approach.

**Limitations And Societal Impact:**

- In Table 1, I am surprised that the proposed method performs very well on the UPT dataset (~7.8 FID, compared to 21.58 in DeepFashion). Other methods show similar performance on the two datasets. It is better to discuss the root cause here. I am a little worried that the proposed design choices are particularly suitable for UPT, but they may not always be very helpful in other datasets.
- It is better to have an apple-to-apple comparison to further demonstrate the advantages of the proposed dataset. Figure 1(b) in the supplement is a good example. I want to know how the 25 clothing categories are distributed in the new dataset compared to the other two. Since the authors have proposed a new dataset, it is better to display detailed statistics to convince readers of the benefits of the dataset.
- Figure 5, 1st row, left example. It seems that the proposed method smoothed out the black belt. Is it because of the error in the predicted garment mask? It's better to discuss.
- It is better to show the running time compared with other methods, because the speed of inference is also a consideration for virtual try-on.
- Minor issue: L40, the main text is too close to the caption of the figure 1.

**Main Review:**

The proposed pose-guided patch-routed disentanglement module is interesting, and the method is technically reasonable. Detailed ablation studies have proven the effectiveness of each design choice. The authors also propose a new dataset that can facilitate further research. Both quantitative and qualitative results are promising. I also appreciate the detailed descriptions in the supplemental material. In general, I am inclined to accept. I have some concerns about the experiment and the dataset, and I want to hear the authors' response to re-evaluate my score.

**Time Spent Reviewing:**

4

---

> ### Author Response · Authors · 2021-08-09
> **Response to Reviewer vykN**
>
> We are pleased to hear that you find the proposed PASTA-GAN interesting and that you appreciate our experiment results, the newly collected dataset, and the detailed descriptions. In the following, we will address your concerns:
>
> * **Discrepancy of the FID score**: "The reason for the discrepancy of the FID score among different datasets and the poor performance for other baseline methods on the UPT dataset."
> * **Comparison of the category distribution**: "Comparison of the distribution of the category among the UPT, MPV, DeepFashion."
> * **Disappearance of the black belt**: "The reason for the disappearance of the black belt in Figure 5, 1st row, 1st column."
> * **Running time**: "Running time compared with other methods."
> * **Minor issue**: "L40, the main text is too close to the caption of the Figure 1."
>
> ## **Discrepancy of the FID score**
>
> We find that there is a negative correlation between the FID score and the amount of data used for the calculation of the FID score. To verify this observation, we conduct an additional experiment. We randomly sample data from the testing set of UPT to form a new dataset for real data and randomly sample data from the synthesis results of PASTA-GAN on UPT with the same sampling rate to form another dataset for fake data. Then, we calculate the FID score between the sampled real data and the sampled fake data. The FID scores for the sampled data with sampling rate of 1.0, 0.9, 0.8, 0.7, 0.6, 0.5, 0.4, 0.3, 0.2, 0.1 are 7.852, 8.198, 8.592, 9.316, 9.833, 10.94, 12.48, 15.20, 20.05, and 30.49, respectively. The results illustrate that when the sampling rate increases, the FID score decreases, which is consistent with our observation. Note that when using the sampling rate of 0.2, the amount of data used for calculation is 1224, which is similar to the amount of data in the DeepFashion test set (i.e., 943), and the FID score (i.e., 20.05) is also close to the FID score for Deepfashion (i.e., 21.58).
> The amount of testing data for UPT, MPV, and DeepFashion are 6115, 2393, and 943, respectively. Thus we argue that it is reasonable for PASTA-GAN to obtain the lowest FID score on the UPT dataset.
>
> Since there are few released works for the unpaired virtual try-on, we can only choose the traditional paired try-on methods like CP-VTON[1], ACGPN[2], PFAFN[3], and some relevant works like ADGAN[4] and liquid warping GAN[5] (which are mainly designed for pose transfer) as baselines. These methods are unable to obtain a satisfactory FID score under the unpaired setting. Therefore, to fairly compare with the paired try-on baselines, we also conduct experiments on the MPV dataset[6], in which different methods are tested under their corresponding setting. Table 2 illustrates the superiority of our PASTA-GAN under such a fair comparison.
>
> ## **Comparison of the category distribution**
>
> It is laborious to label the whole dataset within a week. Thus, for each dataset, we randomly sample 200 images and manually assign the category labels and the gender label for each image. We will try to label the whole dataset in the following days and display the true distribution in the revision. However, we believe that the distribution of the sub-dataset approximates the truth distribution relatively well. For the upper garment, we define six categories, i.e., short sleeve(c1), long sleeve(c2), short blouse(c3), long blouse(c4), vest(c5), sling(c6). For the lower garment, we define four categories, i.e., pants(c7), shorts(c8), skirts(c9), dress(c10). In the following, we denote each garment category by their class name (c1 for short sleeve, c2 for long sleeve, etc.)
>
> For the UPT dataset, the proportion for c1, c2, c3, c4, c5, c6 are 0.492, 0.402, 0.010, 0.030, 0.050, and 0.015, respectively. The proportion for c7, c8, c9, c10 are 0.845, 0.063, ,0.080, and 0.011, respectively. The proportion for female and male are 0.525 and 0.475.
> For the MPV dataset[6], the proportion for c1, c2, c3, c4, c5, c6 are 0.410, 0.360, 0, 0, 0.19, and 0.040, respectively. The proportion for c7, c8, c9, c10 are 0.898, 0.018, 0.084, and 0, respectively. The proportion for female and male are 1 and 0.
> For the DeepFashion dataset[7], the proportion for c1, c2, c3, c4, c5, c6 are 0.29, 0.24, 0.030, 0.010, 0.260, and 0.170, respectively. The proportion for c7, c8, c9, c10 are 0.516, 0.356, 0.128, and 0 respectively. The proportion for female and male are 0.91 and 0.09.
>
> Considering the diversity of the garment categories in the sampled data, the UPT dataset involves all defined categories, while DeepFashion lacks the dress category and MPV lacks the short blouse, long blouse, and dress categories. Although the distribution among different categories is more balanced for the DeepFashion dataset, it is considerably smaller than the UPT dataset. Furthermore, our UPT dataset has a more balanced gender distribution.
>
> ## **Disappearance of the black belt**
>
> The reason for the disappearance of the black belt is a parsing error.  More specifically, the human parsing model[8] that we use does not designate a label for the belt, and the human parsing estimator[9] will therefore assign a label for the belt region (i.e. pants, upper clothes, background, etc). For the particular example displayed in Figure 5, 1st row, 1st column, the parsing label for the belt region is assigned the background label. This means that the pants that are obtained according to the predicted human parsing will not contain the belt, which will therefore also not be contained in the normalized pants patches and the warped pants. The style synthesis branch then predicts the precise mask for the pants (including the belt region) and the texture synthesis branch inpaints the belt region with the white color according to the features of the pants, thus "smoothing out" the belt.
>
> We will mention this in the revised version.
>
> ## **Running time**
>
> We agree with your opinion that running time is a significant factor for the virtual try-on system. We try to fairly compare the running time among our PASTA-GAN and the other baseline methods and find that our PASTA-GAN is superior to most of the baseline methods in terms of the running time. More specifically, the inference time for one try-on process for CP-VTON[3], ACGPN[4], PFAFN[5], ADGAN[6], Liquid Warping GAN[7], and our proposed PASTA-GAN are 0.021s, 0.104s, 0.067s, 0.164s, 84.441s, and 0.018s, respectively. For a fair comparison, all the methods except PFAFN are tested on the same machine using one NVIDIA GeForce RTX 3090 Graphics Card. For PFAFN, we test it on another machine with one GeForce GTX 1660 Ti Graphics Card due to compatibility issues. Note that all of CP-VTON, ACGPN, ADGAN, and our PASTA-GAN rely on the 2D pose and human parsing, which requires an additional 0.190s per image (0.005s for 2D pose estimation and 0.185s for human parsing estimation). Liquid Warping GAN does not take as inputs the 2D pose and human parsing and instead relies on the SMPL which requires much more time for the prediction. Since the official code for Liquid Warping GAN deeply entangles the data pre-processing and model inference, the running time for Liquid Warping GAN mentioned above already includes the time for both processes. Since PFAFN leverages knowledge distillation to train a parser-free student model, there is no extra cost for data pre-processing. Finally, for one try-on process, the total running time for CP-VTON, ACGPN, PFAFN, ADGAN, Liquid Warping GAN, and PASTA-GAN are 0.211s, 0.294s, 0.067s, 84.441s, 0.208s, respectively. We can observe that the main time cost of our PASTA-GAN is from the data pre-processing and that it has a competitive running time compared to most of the existing virtual try-on methods.
>
>
> ## **Minor issue**
>
> Thank you for pointing this out. We will fix this in the revision.
>
> ---
>
> [1] Wang et al. "Toward characteristic-preserving image-based virtual try-on network.", 2018.
>
> [2] Yang et al. "Towards photo-realistic virtual try-on by adaptively generating preserving image content.", 2020.
>
> [3] Ge et al. "Parser-free virtual try-on via distilling appearance flows.", 2021.
>
> [4] Men et al. "Controllable person image synthesis with attribute-decomposed gan.", 2020.
>
> [5] Liu et al. "Liquid warping gan: A unified framework for human motion imitation, appearance transfer and novel view synthesis.", 2019.
>
> [6] Dong et al. "Towards multi-pose guided virtual try-on network.", 2019.
>
> [7] Liu et al. "Deepfashion: Powering robust clothes recognition and retrieval with rich annotations.", 2016.
>
> [8] Liang et al. "Look into Person: Joint Body Parsing & Pose Estimation Network and A New Benchmark.", 2018.
>
> [9] Gong et al. "Graphonomy: Universal Human Parsing via Graph Transfer Learning.", 2019.

---

> > ### Comment · Reviewer_vykN · 2021-08-23
> > **Authors have addressed my concerns**
> >
> > Thanks for the detailed explanation. The author has resolved most of my concerns. As the authors have pointed out, although the distribution among different categories is more balanced for the DeepFashion dataset, the UPT dataset is considerably larger in scale and balanced in gender.I am somewhat convinced by this response, but it would be better if the UPT dataset can be enriched with a balanced distribution of different categories in the future. I suggest that the authors include the content mentioned in the rebuttal in their draft.

---

> > > ### Author Response · Authors · 2021-08-24
> > > **Thanks to Reviewer vykN**
> > >
> > > Thanks for your constructive suggestion. We will include the statistics analysis of the UPT dataset mentioned in the rebuttal in the next version. Besides, we will try to collect more data for different types of garment to balance the distribution of garment categories of UPT dataset.

---

### Official Review · Reviewer_BGd4 · 2021-07-21

**Rating:** 6
**Confidence:** 4

**Summary:**

This paper presents a GAN model for virtual try-on trained from unpaired image data. The main network is based on StyleGAN2. During the training stage, for each image, the extracted 2D skeleton pose is used to convert the parsed garment region into a set of rectified texture patches with the proposed Patch-routed disentanglement Module. These rectified patches are encoded to be a single vector fed into every stage of the generator. The head region and the 2D skeleton pose are encoded to be identity features as input to the generator. Because the detailed texture information is not well preserved in style patches, the generator first synthesize coarse-level image and garment mask. The garment region is further fused in later stage of the generator to synthesize the final output. The model is trained in a reconstruction way without using paired person images. During the inference stage, the garment region is warped by the target pose as input into the texture synthesis stage of the generator. The key idea is its patch-based garment disentangling. The part-wise patch rectification removes pose and shape related information but only keeps low-level texture information. Results are reported on DeepFashion and MPV and self-collected UPT datasets with comparisons to state-of-the-art methods.

**Limitations And Societal Impact:**

Limitations and societal impact are mostly addressed. It'd be better to provide additional results for viewpoint limitation resulted from 2D skeleton pose as mentioned above.

**Main Review:**

Pros:
1) It is an interesting idea of garment disentangling by rectifying the corresponding garment region on each skeleton bone into a standard square patch. The rectification process makes sure the resulting patches are normalized in terms of pose and shape. It is also the key to enable unpaired learning.
2) The StyleGAN2 based generator is well constructed. Having a separate texture synthesis branch with warped garment as input makes sure the generator can synthesis high-quality texture with the aid of spatially-adaptive residual module.
3) It presents impressive visual results. Its high-quality synthesis is also verified with significant improvements in FID scores and user preferences compared to the related works.

Cons:
1) The proposed patch-routed disentanglement module is based on 2D skeleton pose, which may be only feasible to normalize/rectify garment w.r.t. in-plane pose changes. The presented visual results are all about near frontal views, which seem to agree with this observation. The visual results sometimes look 3D implausible, especially when large patterns/logos/designs are present in the garment. In a recent work by Kripasindhu Sarkar et al. (StylePoseGAN), the DensePose is used to extract UV texture map, which is meant to be 3D pose invariant. How would this paper compare with DensePose-based disentangling?
2) In Line 140, it'd be better to explain how to come up with the numbers of valid patches for different kinds of garments.
3) It'd be also desirable to generate higher-resolution results, 512x512.

**Time Spent Reviewing:**

3 hours

---

> ### Author Response · Authors · 2021-08-09
> **Response to Reviewer BGd4**
>
> Thanks for your detailed and constructive feedback. We are glad to hear that you appreciated the underlying idea of our PASTA-GAN to facilitate unpaired learning via garment disentangling, our well-constructed generator, and the experimental results. In the following, we will address all your questions:
>
> * **Comparison to DensePose disentangling**: "How would this paper compare with DensePose-based disentangling?"
> * **Number of patches**: "How to come up with the numbers of valid patches for different kinds of garments?"
> * **Higher resolution results**: "It would be desirable to generate higher-resolution results."
>
>
> ### **Comparison to DensePose disentangling**
>
> The garment patches obtained according to the DensePose[1] UV map can not guarantee the completeness of the source garment, since it can only extract the texture inside the region provided by the DensePose model and neglects the area outside of it. This can cause problems when considering loose clothing. Take as an example the long sleeve garments. If we utilize DensePose, the garment patch around the arm can only contain the garment texture inside the arm region, regardless of how loose the sleeve is. Such incomplete garment patches will result in imprecise warped garments and impair the texture synthesis of the try-on result.
> In our proposed disentangling method, instead, we can account for this by adjusting the garment region through the patch size hyper-parameter. Further, the DensePose is more complicated than our 2D pose, and thus requires a more elaborate model for precise prediction, increasing the computational cost of data pre-processing.
>
> We will explicitly analyze the difference between these two disentangling methods in the revision.
>
> Furthermore, your observation is correct and PASTA-GAN mainly aims to tackle the unpaired virtual try-on for front view garment transfer. Unpaired try-on with large 3D pose change (e.g. transferring a garment onto a side view of a person) is not our focus in this work. However, we believe that it is a valuable research direction and will further explore how to make our model robust to 3D pose changes in future work.
>
>
> ### **Number of patches**
>
> Thank you for pointing out that this might not be clear to the reader. As illustrated in Figure 3, we design eight patches for the upper-body garments, i.e., the patches around the left/right upper/bottom arm (4), the patches around the left/right pelvis (2), a patch around the torso (1), and a patch around the neck (1). The number of valid patches for a particular garment type depends on the number of patches that contain the specific garment. For instance, the number of valid patches for T-shirts will be six because the right and left bottom arm patches will not contain the garment.
>
> We will explain this more clearly in the revised version and provide details about how we define the patches for the upper-body and lower-body garments and the definition of valid patches.
>
>
> ### **Higher resolution results**
>
> While we only present results with a resolution of 256x256 in the paper, this is not a limitation of the PASTA-GAN, which can synthesize higher resolution results. However, as the main focus of this work is on scaleable unpaired virtual try-on, we adapted the same resolution as our main baselines (i.e. ADGAN[2], Liquid Warping GAN[3]) for fair comparisons.
>
> We will display additional try-on results with a resolution of 512x512 in the revision.
>
> ---
>
> [1] Güler et al. "DensePose: Dense Human Pose Estimation In The Wild.", 2018.
>
> [2] Men et al. "Controllable person image synthesis with attribute-decomposed gan.", 2020.
>
> [3] Liu et al. "Liquid warping gan: A unified framework for human motion imitation, appearance transfer and novel view synthesis.", 2019.

---

### Decision · Program_Chairs · 2021-09-27

**Decision:**

Accept (Poster)

**Comment:**

The paper was reviewed by four expert reviewers in the community. Most reviewers appreciate the novelty on unpaired training for virtual try-on (although paired data is not a critical restriction in this problem domain). The conditional extension of StyleGAN2 is somewhat similar to StylePoseGAN. All reviewers are aware that StylePoseGAN is an unpublished work so the reviewer assessment of this paper is not affected by an arXiv paper. There were an extensive discussions between the authors and Reviewer 5Zj3. The AC reads the reviews, authors' rebuttal, and the discussions. While there are still some clarifications required for the method exposition and several other limiting factors (e.g., low resolution results only, similar approach as concurrent work), the AC thinks the paper has sufficient merits and could inspire future research work. The AC thus recommends to accept.